# INFORMATION-THEORETIC LOCAL MINIMA CHARACTERIZATION AND REGULARIZATION

## ABSTRACT

Recent advances in deep learning theory have evoked the study of generalizability across different local minima of deep neural networks (DNNs). While current work focused on either discovering properties of good local minima or developing regularization techniques to induce good local minima, no approach exists that can tackle both problems. We achieve these two goals successfully in a unified manner. Specifically, based on the Fisher information we propose a metric both strongly indicative of generalizability of local minima and effectively applied as a practical regularizer. We provide theoretical analysis including a generalization bound and empirically demonstrate the success of our approach in both capturing and improving the generalizability of DNNs. Experiments are performed on CIFAR-10 and CIFAR-100 for various network architectures.

## 1 INTRODUCTION

Recently, there has been a surge in the interest of acquiring a theoretical understanding over deep neural network's behavior. Breakthroughs have been made in characterizing the optimization process, showing that learning algorithms such as stochastic gradient descent (SGD) tend to end up in one of the many local minima which have close-to-zero training loss (Choromanska et al., 2015; Dauphin et al., 2014; Kawaguchi, 2016; Nguyen & Hein, 2018; Du et al., 2018). However, these numerically similar local minima typically exhibit very different behaviors in terms of generalizability. It is, therefore, natural to ask two closely related questions: (a) What kind of local minima can generalize better? (b) How to find those better local minima?

To our knowledge, existing work focused only on one of the two questions. For the "what" question, various definitions of "flatness/sharpness" have been introduced and analyzed (Keskar et al., 2017; Neyshabur et al., 2018; 2017; Wu et al., 2017; Liang et al., 2017). However, they suffer from one or more of the problems: (1) being mostly theoretical with no or poor empirical evaluations on modern neural networks, (2) lack of theoretical analysis and understanding, (3) in practice not applicable to finding better local minima. Regarding the "how" question, existing approaches (Hochreiter & Schmidhuber, 1997; Sokolić et al., 2017; Chaudhari et al., 2017; Hoffer et al., 2017; Neyshabur et al., 2015a; Izmailov et al., 2018) share some of the common drawbacks: (1) derived only from intuitions but no specific metrics provided to characterize local minima, (2) no or weak analysis of such metrics, (3) not applicable or no consistent generalization improvement for modern DNNs.

In this paper, we tackle both the "what" and the "how" questions in a unified manner. Our answer provides both the theory and applications for the generalization problems across different local minima. Based on the determinant of Fisher information estimated from the training set, we propose a metric that *solves all the aforementioned issues*. The metric can well capture properties that characterize local minima of different generalization ability. We provide its theoretical analysis, primarily a generalization bound based on PAC-Bayes (McAllester, 1999b;a). For modern DNNs in practice, it is necessary to provide a tractable approximation of our metric. We propose an intuitive and efficient approximation to compare it across different local minima. Our empirical evaluations fully illustrate the effectiveness of the metric as a strong indicator of local minima's generalizability. Moreover, from the metric we further derive and design a practical regularization technique that guides the optimization process in finding better generalizable local minima. The experiments on image classification datasets demonstrate that our approach gives consistent generalization boost for a range of DNN architectures.

## 2 RELATED WORK

It has been empirically shown that larger batch sizes lead to worse generalization (Keskar et al., 2017). Hoffer et al. (2017) analyzed how the training dynamics is affected by different batch sizes and presented a perturbed batch normalization technique for better generalization. While it effectively improves generalization for large-batch training, a specific metric that indicates the generalizability is missing. Similarly, Elsayed et al. (2018) employed a structured margin loss to improve performance of DNNs w.r.t. noise and adversarial attack yet no metric was proposed. Furthermore, this approach essentially provided no generalization gain in the normal training setup.

The local entropy of the loss landscape was proposed to measure "flatness" in Chaudhari et al. (2017), which also designed an entropy-guided SGD that achieves faster convergence in training DNNs. However, the method does not consistently improve generalization, e.g., a decrease of performance on CIFAR-10 (Krizhevsky & Hinton, 2009). Another method that focused on modifying the optimization process is the Path-SGD proposed by Neyshabur et al. (2015a). Specifically, the authors derived an approximate steepest descent algorithm that utilizes the path-wise norm regularization to achieve better generalization. The authors only evaluated it on a two-layer neural network, very likely since the path norm is computationally expensive to optimize during training.

A flat minimum search algorithm was proposed by Hochreiter & Schmidhuber (1997) based on the "flatness" of local minima defined as the volume of local boxes. Yet since the boxes have their axes aligned to the axes of the model parameters, their volumes could be significant underestimations of "flatness" for over-parametrized networks, due to the specific spectral density of Hessian of DNNs studied in Pennington & Worah (2018); Sagun et al. (2018). The authors of Wu et al. (2017) also characterized the "flatness" by volumes. They considered the inverse volume of the basin of attraction and proposed to use the Frobenius norm of Hessian at the local minimum as a metric. In our experiments, we show that their metric does not accurately capture the generalization ability of local minima under different scenarios. Moreover, they have not derived a regularizer from their metric.

Based on a "robustness" metric, Sokolić et al. (2017) derived a regularization technique that successfully improves generalization on multiple image classification datasets. Nevertheless, we show that their metric fails to capture the generalizability across different local minima.

By using the Bayes factor, MacKay (1992) studied the generalization ability of different local minima obtained by varying the coefficient of L2 regularization. It derived a formula involving the determinant of Hessian, similar to the one in ours. Whereas, this approach has restricted settings and, without proposing an efficient approximation, its metric is not applicable to modern DNNs, let alone serving as a regularizer. A generalization bound is missing in MacKay (1992) as well.

In a broader context of the "what" question, properties that capture the generalization of neural networks have been extensively studied. Various complexity measures for DNNs have been proposed based on norm, margin, Lipschitz constant, compression and robustness (Bartlett & Mendelson, 2002; Neyshabur et al., 2015b; Sokolić et al., 2017; Xu & Mannor, 2012; Bartlett et al., 2017; Zhou et al., 2019; Dziugaite & Roy, 2017; Arora et al., 2018; Jiang et al., 2019). While some of them aimed to provide tight generalization bounds and some of them to provide better empirical results, none of the above approaches explored the "how" question at the same time.

Very recently, Karakida et al. (2019) and Sun & Nielsen (2019) studied the Fisher information of the neural network through the lens of its spectral density. In specific, Karakida et al. (2019) applied mean field theory to study the statistics of the spectrum and the appropriate size of the learning rate. Also an information-theoretic approach, Sun & Nielsen (2019) derived a novel formulation of the minimum description length in the context of deep learning by utilizing tools from singular semi-Riemannian geometry.

## 3 OUTLINE AND NOTATIONS

In a typical $K$-way classification setting, each sample $x \in \mathcal{X}$ belongs to a single class denoted $c_x \in \{0, 1, ..., K\}$ according to the probability vector $y \in \mathcal{Y}$, where $\mathcal{Y}$ is the k-dimensional probability simplex so that $p(c_x = i) = y_i$ and $\sum_i y_i = 1$. Denote a feed-forward DNN parametrized by $w \in \mathbb{R}^W$ as $f_w : \mathcal{X} \to \mathcal{Y}$, which uses nonlinear activation functions and a softmax layer at the end. Denote the cross entropy loss as $\ell(f_w(x), y) = -\sum_i y_i \ln f_w(x)_i$. De-

note the training set as $\mathcal{S}$, defined over $\mathcal{X} \times \mathcal{Y}$ with $|\mathcal{S}| = N$. The training objective is given as $\mathcal{L}(\mathcal{S}, w) = \frac{1}{N} \sum_{(x,y) \sim \mathcal{S}} \ell(f_w(x), y)$. Assume $\mathcal{S}$ is sampled from some true data distribution denoted $\mathcal{D}$, we can define expected loss $\mathcal{L}(\mathcal{D}, w) = \mathbb{E}_{(x,y) \sim \mathcal{D}}[\ell(f_w(x), y)]$. Throughout this paper, we refer a local minimum of $\mathcal{L}(\mathcal{S}, w)$ corresponding to a local minimizer $w_0$ as just the local minimum $w_0$. Given such $w_0$, our paper's outline as well as our main achievements are:

- In Section 4 we relates Fisher information to neural network training as a prerequisite.
- In Section 5.1 we propose a metric $\gamma(w_0)$ that well captures local minima's generalizability.
- In Section 5.2 we provide a generalization bound related to $\gamma(w_0)$.
- In Section 5.3 we propose an approximation $\widehat{\gamma}(w_0)$ for $\gamma(w_0)$, which is shown to be very effective in Section 7.1 via extensive empirical evaluations.
- In Section 6 we devise a practical regularizer from $\gamma(w_0)$ that consistently improves generalizability across different DNNs, as evaluated in Section 7.2.

## 3.1 OTHER NOTATIONS

Denote $\nabla_w$ as gradient, $\mathbf{J}_w[\cdot]$ as Jacobian matrix, $\nabla_w^2$ as Hessian, $D_{\mathrm{KL}}(\cdot\|\cdot)$ as KL divergence, $\|\cdot\|_2$ as spectrum or Euclidean norm, $\|\cdot\|_F$ as Frobenius norm, $|\cdot|$ as determinant, $\mathrm{tr}(\cdot)$ as trace norm, $\rho(\cdot)$ as spectral radius, $\ell\ell_\mathcal{S}(w)$ as log-likelihood on $\mathcal{S}$, and $[\cdot]_i$ for selecting the $i^{\mathrm{th}}$ entry.

We define $\boldsymbol{\ell}_x(w) \in \mathbb{R}^K$ whose $i^{\mathrm{th}}$ entry is $-\ln f_w(x)_i$ so that $\ell(f_w(x), y) = \boldsymbol{\ell}_x(w)^T y$. We define $\bar{y}$ as $\mathrm{argmax}(y)$ and $\tilde{y} \in \mathbb{R}^K$ the one-hot vector whose $\bar{y}$-th entry is 1 and otherwise 0. Then we define $\tilde{\mathcal{L}}(\mathcal{S}, w) \in \mathbb{R}^N$ as the "simplified" loss vector of $\mathcal{S}$ whose entries are $\ell(f_w(x), \tilde{y})$ for $(x, y) \in \mathcal{S}$, i.e., we approximate the cross entropy loss $\ell(f_w(x), y)$ by $\ell(f_w(x), \tilde{y})$.

## 4 LOCAL MINIMUM AND FISHER INFORMATION

First of all, if $y$ is strictly one-hot, no local minimum will even exist with 100% training accuracy, since the cross entropy loss will always be positive. To admit good local minima in the first place, we assume the widely used label smoothing (Szegedy et al., 2016) is applied to train all models in our analysis. Label smoothing enables us to assume a local minimum $w_0$ (in this case, also a global minimum) of the training loss with $\sum_{(x,y) \in \mathcal{S}} D_{\mathrm{KL}}(f_{w_0}(x)\|y) = 0$.

Each sample $(x, y) \in \mathcal{S}$ has its label $c_x$ sampled by $p(c_x = i|x) = y_i$, denoted as $(x, c_x) \sim \mathcal{S}$. The joint probability $p(x, c_x)$ modeled by the DNN is $p(x, c_x = i; w) = p(c_x = i|x; w) \, p(x) = [f_w(x)]_i \, p(x)$ with $p(x) = \frac{1}{N}$. We can relate the training loss $\mathcal{L}(\mathcal{S}, w)$ to the negative log-likelihood $-\ell\ell_\mathcal{S}(w) = -\sum_{(x,y) \in \mathcal{S}} \mathbb{E}_{c_x \sim y} \ln p(x, c_x; w)$ by:

$$\mathcal{L}(\mathcal{S}, w) = \frac{1}{N} \sum_{(x,y) \in \mathcal{S}} \boldsymbol{\ell}_x(w)^T y = -\frac{1}{N} \sum_{(x,y) \in \mathcal{S}} \mathbb{E}_{c_x \sim y} \ln p(c_x|x; w) = -\frac{1}{N} \ell\ell_\mathcal{S}(w) + \ln \frac{1}{N}$$

And so $w_0$ also corresponds to a local maximum of the likelihood function. The observed Fisher information evaluated at $w_0$ is defined as $\mathcal{I}(w_0) = -\frac{1}{N} \nabla_w^2 \ell\ell_\mathcal{S}(w_0)$. We can further derive:

$$\mathcal{I}(w_0) = \nabla_w^2 \mathcal{L}(\mathcal{S}, w_0) = \mathbb{E}_{(x,c_x) \sim \mathcal{S}} [\nabla_w \ln p(c_x|x; w_0) \nabla_w \ln p(c_x|x; w_0)^T] \quad (1)$$

The first equality is straightforward; the second has its proof in Appendix A. Since $p(c_x = i|x) = y_i$ and $\ln p(c_x = i|x; w_0) = [\boldsymbol{\ell}_x(w_0)]_i$, we can further simplify the Equation 1 to:

$$\mathcal{I}(w_0) = \frac{1}{N} \sum_{(x,y) \in \mathcal{S}} \sum_{i=1}^K \left( \nabla_w [\boldsymbol{\ell}_x(w_0)]_i \right) \left( \nabla_w [\boldsymbol{\ell}_x(w_0)]_i \right)^T \quad (2)$$

**Remark**: When we assume global optimality, we have $\nabla_w \ell(f_{w_0}(x), y) = \mathbf{0}$ as $D_{\mathrm{KL}}(f_{w_0}(x)\|y) = 0$; yet it does not indicate $\mathcal{I}(w_0) \in \mathbb{R}^{W \times W} \neq \mathbf{0}$ in Equation 2.

## 5 LOCAL MINIMA CHARACTERIZATION

In this section, we derive and propose our metric, provide a PAC-Bayes generalization bound, and lastly, propose and give intuitions of an effective approximation of our metric for modern DNNs.

## 5.1 FISHER DETERMINANT AS GENERALIZATION METRIC

We would like a metric to compare different local minima. Similar to the various definitions of "flatness/sharpness", we take a small neighborhood of the target local minimum $w_0$ into account. Formally for a sufficiently small $V$, we define the model class $\mathcal{M}(w_0)$ as the largest connected subset of $\{w \in \mathbb{R}^W : \mathcal{L}(\mathcal{S}, w) \leq h\}$ that contains $w_0$, where the height $h$ is defined as a real number such that the volume (namely the Lebesgue measure) of $\mathcal{M}(w_0)$ is $V$. By the Intermediate Value Theorem, for any sufficiently small volume $V$ there exists a corresponding height $h$.

We propose our metric $\gamma(\cdot)$, where lower $\gamma(w_0)$ indicates a better local minimum $w_0$:

$$\gamma(w_0) = \ln|\mathcal{I}(w_0)| \tag{3}$$

As a metric, $\gamma(w_0)$ requires $|\mathcal{I}(w_0)| \neq 0$. Therefore, we state the following Assumption 1.

**Assumption 1.** *The local minima $w_0$ we care about in the comparison are well isolated and unique in their corresponding neighborhood $\mathcal{M}(w_0)$.*

The Assumption 1 is quite reasonable. For state-of-the-art network architectures used in practice, this is often the fact. To be precise, the Assumption 1 is violated when the Hessian matrix at a local minimum is singular. Specifically, Orhan & Pitkow (2018) summarizes three sources of the singularity: (i) due to a dead neuron, (ii) due to identical neurons, and (iii) linear dependence of the neurons. As well demonstrated in Orhan & Pitkow (2018), network with skip connection, e.g. ResNet (He et al., 2016a), WRN (Zagoruyko & Komodakis, 2016), and DenseNet (Huang et al., 2017) used in our experiments, can effectively eliminate all the aforementioned singularity.

In Dinh et al. (2017), the authors pointed out another source of the singularity specifically for networks with scale-invariant activation functions, e.g. ReLU, referred as the rescaling issue. Namely, one can rescales the model parameters layer-wise so that the underlying function represented by the network remains unchanged in the region. In practice, this issue is not critical. Firstly, most modern deep ReLU networks, e.g. ResNet, WRN, and DenseNet, have normalization layers, e.g. BatchNorm (Ioffe & Szegedy, 2015), applied before the activations. BatchNorm shifts all the inputs to the ReLU function, equivalently shifting the ReLU horizontally which makes it no longer scale-invariant. Secondly, due to the ubiquitous use of Gaussian weights initialization scheme and weight decay, most local minima obtained by gradient learning have weights of a relatively small norm. Consequently, in practice, we will not compare two local minima essentially the same but have one as the rescaled version of the other with a much larger norm of the weights.

Note that normally we have a limited size of the dataset, and so an approximation of $\gamma(w_0)$ is a must. We present our approximation scheme and its intuition in Section 5.3.

### 5.1.1 CONNECTION TO FISHER INFORMATION APPROXIMATION (FIA) CRITERION

Our metric $\gamma(w_0)$ is closely related to the FIA criterion. From Information Theory, the MDL principle suggests that among different statistical models the best is the one that best compresses both the sampled data and the model (Rissanen, 1978). Accordingly, Rissanen (1996) derived the FIA criterion to compare statistical models, each of which is a class of model in the neighborhood of a global minimum $w_0$. The model class's FIA criterion is written as (lower FIA is better):

$$\text{FIA} = - \sum_{(x,y) \in \mathcal{S}} \mathop{\mathbb{E}}_{c_x \sim y} \ln p(x, c_x; w_0) + \frac{W}{2} \ln \frac{N}{2\pi} + \ln \int_{\mathcal{M}(w_0)} \sqrt{|\mathcal{I}_w|} \, dw$$

On the right hand side, the first two terms are both constants. To see the connection to our metric, we replace the expected Fisher information $\mathcal{I}_w$ with the tractable observed one $\mathcal{I}(w_0)$. Assuming the training loss is locally quadratic in $\mathcal{M}(w_0)$, an assumption later formalized and validated as Assumption 2, since $\mathcal{I}(w_0) = \nabla_w^2 \mathcal{L}(\mathcal{S}, w_0)$ from Equation 1, we can modify the last term to be $\ln V + \ln \sqrt{|\mathcal{I}(w_0)|}$.

**Remark**: Although in a similar format, the FIA criterion and our metric are essentially different due to the appearance of observed Fisher information in place of the expected one, making our metric both tractable and much more applicable (no longer requires global optimality).

### 5.1.2 CONNECTION TO EXISTING FLATNESS/SHARPNESS METRICS

As mentioned in Section 2, the "flatness" of a local minimum was firstly related to the generalization ability of the neural network in Hochreiter & Schmidhuber (1997), where the concept and the method are both preliminary. The idea is recently popularized in the context of deep learning by a series of paper such as Keskar et al. (2017); Chaudhari et al. (2017); Wu et al. (2017). Our approach roughly shares the same intuition with these existing works, namely, a "flat" local minimum admits less complexity and so generalizes better than a "sharp" one. To our best knowledge, our paper is the first among these work that provides both the theoretical analysis including a generalization bound and the empirical verification of both an efficient metric and a practical regularizer for modern network architectures.

## 5.2 GENERALIZATION BOUND

**Assumption 2.** *Given the training loss $\mathcal{L}(\mathcal{S}, w)$, its local minimum $w_0$ satisfying Assumption 1 and the associated neighborhood $\mathcal{M}(w_0)$ whose volume $V$ is sufficiently small, as described in Section 3, 4 and 5.1, respectively, when confined to $\mathcal{M}(w_0)$, we assume that $\mathcal{L}(\mathcal{S}, w)$ is quadratic.*

The Assumption 2 is quite reasonable as well. Grünwald & Grunwald (2007) suggests that, a log-likelihood function, under regularity conditions (1) existence of its $1^{\text{st}}$, $2^{\text{nd}}$ & $3^{\text{rd}}$ derivatives and (2) uniqueness of its maximum in the region, behaves locally like a quadratic function around its maximum. In our case, $\mathcal{L}(\mathcal{S}, w)$ corresponds to the log-likelihood function $\ell\ell_{\mathcal{S}}(w)$ and so $w_0$ corresponds to a local maximum of $\ell\ell_{\mathcal{S}}(w)$. Since $\mathcal{L}(\mathcal{S}, w)$ is analytic and $w_0$ is the only local minimum of $\mathcal{L}(\mathcal{S}, w)$ in $\mathcal{M}(w_0)$, the training loss indeed can be considered locally quadratic.

Similar to Langford & Caruana (2002), Harvey et al. (2017) and Neyshabur et al. (2017), we apply the PAC-Bayes Theorem (McAllester, 2003) to derive a generalization bound for our metric. Specifically, we pick a uniform prior $\mathcal{P}$ over $w \in \mathcal{M}(w_0)$ according to the maximum entropy principle and after observing the training data $\mathcal{S}$ pick the posterior $\mathcal{Q}$ of density $q(w) \propto e^{-|\mathcal{L}(\mathcal{S}, w_0) - \mathcal{L}(\mathcal{S}, w)|}$. Then Theorem 1 bounds the expected $\mathcal{L}(\mathcal{D}, w)$ using $\gamma(w_0)$. See its proof in Appendix B.

**Theorem 1.** *Given $|\mathcal{S}| = N$, $\mathcal{D}$, $\mathcal{L}(\mathcal{S}, w)$ and $\mathcal{L}(\mathcal{D}, w)$ described in Section 3, a local minimum $w_0$ with $\mathcal{L}_0 \triangleq \mathcal{L}(\mathcal{S}, w_0)$, the volume $V$ of $\mathcal{M}(w_0)$ sufficiently small, the Assumption 1 & 2 satisfied, and $\mathcal{P}, \mathcal{Q}$ defined above, for any $\delta \in (0, 1]$, we have with probability at least $1 - \delta$ that:*

$$\mathbb{E}_{w \sim \mathcal{Q}}[\mathcal{L}(\mathcal{D}, w)] \leq \mathcal{L}_0 + \mathcal{A} + 2\sqrt{\frac{2\mathcal{L}_0 + 2\mathcal{A} + \ln \frac{2N}{\delta}}{N - 1}}, \quad \mathcal{A} = \frac{WV^{\frac{2}{W}} \pi^{\frac{1}{W}} e^{\gamma(w_0)/W}}{4\pi e}$$

In short, Theorem 1 shows that a lower $\gamma(w_0)$ indicates a more generalizable local minimum $w_0$.

## 5.3 APPROXIMATION

As stated in Section 4, in practice an approximation of $\gamma(w_0)$ as $\widehat{\gamma}(w_0)$ is necessary, as calculating $\gamma(w_0)$ involves computing the determinant of a $W \times W$ matrix. Let us first assume we have an imagined training set $\mathcal{S}'$ of size $W$, a local minimum $w_0$ of $\mathcal{L}(\mathcal{S}', w)$ and so correspondingly a full-rank observed Fisher information matrix $\mathcal{I}'(w_0)$ so that $\ln|\mathcal{I}'(w_0)|$ is well defined. In reality, we only have a training set $\mathcal{S} \subset \mathcal{S}'$ with a singular $\mathcal{I}(w_0)$. Notice that $w_0$ is also a local minimum of $\mathcal{L}(\mathcal{S}, w)$ since $\sum_{(x,y) \in \mathcal{S}'} D_{\text{KL}}(f_{w_0}(x) \| y) = 0$ as assumed in Section 4. We then approximate eigenvalues of $\mathcal{I}'(w_0)$ by those of its sub-matrices and so to approximate $\ln|\mathcal{I}'(w_0)|$.

First of all, we replace $y$ by its one-hot version $\tilde{y}$ defined in Section 3.1 since they are very close. This drastically reduces the cost of gradient calculation. With $\tilde{\mathcal{L}}(\mathcal{S}, w) \in \mathbb{R}^N$ and $\bar{y}$ defined in Section 3.1, according to Equation 2, the observed Fisher information $\mathcal{I}'(w_0) \in \mathbb{R}^{W \times W}$ is:

$$\mathcal{I}'(w_0) \approx \frac{1}{W} \sum_{(x,y) \in \mathcal{S}'} \left( \nabla_w [\ell_x(w_0)]_{\bar{y}} \right) \left( \nabla_w [\ell_x(w_0)]_{\bar{y}} \right)^T$$

$$= \frac{1}{W} \mathbf{J}_w[\tilde{\mathcal{L}}(\mathcal{S}', w)]^T \mathbf{J}_w[\tilde{\mathcal{L}}(\mathcal{S}', w)] = \frac{1}{W} \mathbf{J}_w[\tilde{\mathcal{L}}(\mathcal{S}', w)] \mathbf{J}_w[\tilde{\mathcal{L}}(\mathcal{S}', w)]^T \quad (4)$$

Let $\{\lambda_m\}_{m=1}^W$ denote the eigenvalues of $\mathcal{I}'(w_0)$; then $\gamma(w_0) = \ln \prod_{m=1}^W \lambda_m = \sum_{m=1}^W \ln \lambda_m$. Without calculating all $W$ eigenvalues, we can perform a Monte-Carlo estimation of $\widehat{\gamma}(w_0)$ by

randomly sampling $N' < N < W$ eigenvalues from $\{\lambda_m\}_{m=1}^W$. We denote the samples as $\{\lambda_n\}_{n=1}^{N'}$ and we have $\frac{W}{N'}\sum_{n=1}^{N'}\ln\lambda_n \approx \sum_{m=1}^{W}\ln\lambda_m$. Suppose the estimation is run $T$ times, we have $\lim_{T\to\infty}\frac{1}{T}\sum_{t=1}^{T}\frac{W}{N'}\sum_{n=1}^{N'}\ln\lambda_n = \gamma(w_0)$.

In practice $\{\lambda_n\}_{n=1}^{N'}$ is inaccessible since we don't have $\mathcal{I}'(w_0)$ in the first place. Instead, we sample $\mathcal{S}^t \subset \mathcal{S}$ with $|\mathcal{S}^t| = N'$ for $T$ times and define

$$\xi^t(w_0) \triangleq \mathbf{J}_w[\tilde{\mathcal{L}}(\mathcal{S}^t, w_0)]\mathbf{J}_w[\tilde{\mathcal{L}}(\mathcal{S}^t, w_0)]^T \in \mathbb{R}^{N'\times N'}$$

Notice that $\xi^t(w_0)$ is a principal sub-matrix of $W\mathcal{I}'(w_0)$ by removing rows & columns for data in $\mathcal{S} \setminus \mathcal{S}^t$. According to Theorem 3, one can roughly estimate the size of eigenvalues of a matrix by those of its sub-matrices. Therefore we propose to estimate $\gamma(w_0)$ by $\widehat{\gamma}(w_0)$ with:

$$\widehat{\gamma}(w_0) \triangleq \frac{1}{T}\sum_{t=1}^{T}\ln|\xi^t(w_0)|, \quad \gamma(w_0) \approx \frac{W}{N'}\widehat{\gamma}(w_0) + W\ln\frac{1}{W} \text{ as } T\to\infty \quad (5)$$

We leave Theorem 3 as well as the derivation of Equation 5 to Appendix C. In proposing $\widehat{\gamma}(w_0)$, we ignore the constants and irrelevant scaling factors because what matters is the relative size of $\gamma(\cdot)$ when comparing different local minima. Empirically we find that given relatively large number of sample trials $T$, our metric $\widehat{\gamma}(\cdot)$ can effectively capture the generalizability of a local minimum even for a small $N'$ (details in Section 7.1 and in Appendix D).

## 6  LOCAL MINIMA REGULARIZATION

Besides pragmatism, devising a practical regularizer based on $\gamma(w_0)$ also "verifies" our theoretical understanding of DNN training, helping for future improvement of the learning algorithms. However, converting $\gamma(w_0)$ to a practical regularizer is non-trivial due to the computation burden of:

1. optimizing terms related to the gradient, which involves calculating the Hessian
2. computing the eigenvalues in each training step, which is even more expensive

We first solve the second issue and then the first one. To solve the second issue, we propose to optimize a surrogate term for $\gamma(w_0)$ which avoids eigenvalue computations, namely the trace norm of the observed Fisher information $\text{tr}(\mathcal{I}(w_0))$. These two terms have the relation:

$$\frac{1}{W}\gamma(w_0) = \frac{1}{W}\ln|\mathcal{I}(w_0)| \leq \ln\text{tr}(\mathcal{I}(w_0)) - \ln W$$

Another major benefit of using the trace norm is that, unlike $\gamma(w_0)$, $\text{tr}(\mathcal{I}(w_0))$ still remains well defined even with a small training set $|\mathcal{S}| = N$. From Equation 2 we have:

$$\text{tr}(\mathcal{I}(w_0)) = \frac{1}{N}\sum_{(x,y)\in\mathcal{S}}\sum_{i=1}^{K}\|\nabla_w[\boldsymbol{\ell}_x(w_0)]_i\|_2^2$$

The cost of computing $\text{tr}(\mathcal{I}(w_0))$ is linear in the number of its terms (in the double summation). We therefore simplify the calculation by replacing $y$ with $\tilde{y}$ similar to Equation 4 so that

$$\text{tr}(\mathcal{I}(w_0)) \approx \frac{1}{N}\sum_{(x,y)\in\mathcal{S}}\|\nabla_w\ell(f_{w_0}(x), \tilde{y})\|_2^2 = \frac{1}{N}\sum_{i=1}^{N}\|\nabla_w[\tilde{\mathcal{L}}(\mathcal{S}, w_0)]_i\|_2^2$$

where $\tilde{y}$ and $\tilde{\mathcal{L}}(\cdot, \cdot)$ are defined in Section 3.1. As in gradient-based training we never exactly reach the local minimum $w_0$, we choose to optimize $\text{tr}(\mathcal{I}(w))$ during the entire training process. We have $\nabla_w\frac{1}{|\mathcal{B}|}\sum_i\|\nabla_w[\tilde{\mathcal{L}}(\mathcal{B}, w_0)]_i\|_2^2 = \frac{1}{|\mathcal{B}|}\sum_i\nabla_w\|\nabla_w[\tilde{\mathcal{L}}(\mathcal{B}, w_0)]_i\|_2^2$ for each mini-batch $\mathcal{B}$. Then we can further reduce the computation cost by batching. In specific, we randomly split $\mathcal{B}$ into $M$ sub-batches of equal size, namely $\{\mathcal{B}_i\}_{i=1}^M$. We define $\boldsymbol{g}_i \triangleq \frac{1}{|\mathcal{B}_i|}\sum_{(x,y)\in\mathcal{B}_i}\ell(f_{w_0}(x), \tilde{y})$ and compute $\|\boldsymbol{g}_i\|_2^2$ for $\boldsymbol{g}_i \in \{\boldsymbol{g}_i\}_{i=1}^M$ instead of computing $\|[\tilde{\mathcal{L}}(\mathcal{B}, w_0)]_i\|_2^2$ for each data point in $\mathcal{B}$.

We deal with the first computation burden by adopting first order approximation. For any $w$, with a sufficiently small $\alpha > 0$ we have $\tilde{\mathcal{L}}(\mathcal{B}_i, w - \alpha \boldsymbol{g}_i) \approx \tilde{\mathcal{L}}(\mathcal{B}_i, w) - \mathbf{J}_w[\tilde{\mathcal{L}}(\mathcal{B}_i, w)] \, \alpha \boldsymbol{g}_i$. Then

$$\frac{1}{|\mathcal{B}_i|} \sum_{j=1}^{|\mathcal{B}_i|} [\tilde{\mathcal{L}}(\mathcal{B}_i, w) - \tilde{\mathcal{L}}(\mathcal{B}_i, w - \alpha \boldsymbol{g}_i)]_j \approx \frac{1}{|\mathcal{B}_i|} \sum_{j=1}^{|\mathcal{B}_i|} \left[ \mathbf{J}_w[\tilde{\mathcal{L}}(\mathcal{B}_i, w)] \, \alpha \boldsymbol{g}_i \right]_j = \alpha \|\boldsymbol{g}_i\|_2^2$$

Therefore, we propose to optimize the following regularized training objective for each update step:

$$\mathcal{L}(\mathcal{B}, w) + \beta \mathcal{R}_\alpha(w), \quad \mathcal{R}_\alpha(w) \triangleq \frac{1}{M} \sum_{i=1}^{M} \frac{1}{|\mathcal{B}_i|} \sum_{j=1}^{|\mathcal{B}_i|} \left[ \tilde{\mathcal{L}}(\mathcal{B}_i, w) - \tilde{\mathcal{L}}(\mathcal{B}_i, w - \alpha \boldsymbol{g}_i) \right]_j \tag{6}$$

We omit any second order term when computing $\nabla_w \mathcal{R}_\alpha(w)$, simply by no back-prop through $\boldsymbol{g}_i$. On the other hand, we find that gradient clipping, especially at the beginning of the training, is necessary to make the generalization boost consistent. We have 4 hyper-parameters: $\alpha$, $\beta$, the number of sub-batches $M$ and the gradient clip threshold $\tau$. Our approach is formalized as:

---

**Algorithm 1** Regularized Mini-batch Learning (Single Update Step)

---

1: **procedure** UPDATE($w, \mathcal{B}; \alpha, \beta, M, \tau$)                ▷ Last 4 are hyper-parameters
2:       $\{\mathcal{B}_i\}_{i=1}^M \leftarrow \mathcal{B}$              ▷ Split the mini-batch $\mathcal{B}$ into $M$ sub-batches
3:       **for** $i \leftarrow 1$ to $M$ **do**
4:            $\boldsymbol{g}_i \leftarrow \frac{1}{|\mathcal{B}_i|} \sum_{(x,y) \in \mathcal{B}_i} \ell(f_{w_0}(x), \tilde{y})$     ▷ Compute the gradient of the sub-batch
5:            $\boldsymbol{g}_i = \text{copy}(\boldsymbol{g}_i)$          ▷ A copy prevents gradient flow
6:       **end for**
7:       Compute $\mathcal{R}_\alpha(w)$ by Equation 6          ▷ Use the copied version of $\boldsymbol{g}_i$
8:       $\nabla_w \mathcal{L}_{\text{reg}} \leftarrow \nabla_w[\mathcal{L}(\mathcal{B}, w) + \beta \mathcal{R}_\alpha(w)]$
9:       Clip $\nabla_w \mathcal{L}_{\text{reg}}$ with threshold $\tau$        ▷ `clip_by_global_norm` in TensorFlow
10:      Gradient update with clipped $\nabla_w \mathcal{L}_{\text{reg}}$       ▷ Update with any gradient-based optimizer
11: **end procedure**

---

## 7 EXPERIMENTS

We perform two sets of experiments to illustrate the effectiveness of our metric $\gamma(w_0)$. We demonstrate that: (1) the approximation $\widehat{\gamma}(w_0)$ captures the generalizability well across local minima; (2) our regularization technique based on $\gamma(w_0)$ provides consistent generalization gain for DNNs.

Throughout our theoretical analysis, we assume that label smoothing (LS) is applied during model training in order to obtain well-defined local minima (first mentioned in Section 4). In all our empirical evaluations, we perform both the version with LS applied and without. Results are very similar and so we stick to the version without LS since this is the same as the original training setup in papers of the various network architectures that we used.

### 7.1 EXPERIMENTS ON LOCAL MINIMA CHARACTERIZATION

We perform comprehensive evaluations to compare our metric $\widehat{\gamma}(\cdot)$ with several others on ResNet-20 (He et al., 2016a) for the CIFAR-10 dataset (architecture details in Appendix E). Our metric consistently outperforms others in indicating local minima's generalizability. Specifically, Sokolić et al. (2017) proposed a robustness-based metric used as a regularizer; Wu et al. (2017) proposed to use Frobenius norm of the Hessian as a metric; Keskar et al. (2017) proposed a metric closely related to the spectral radius of Hessian. In summary, we compare 4 metrics, all evaluated at a local minimum $w$ given $\mathcal{S}$. All four metrics go for "smaller values indicate better generalization".

- Robustness: $\frac{1}{N} \sum_{(x,y) \in \mathcal{S}} \|\mathbf{J}_x[f_w(x)]\|_2^2$

- Frobenius norm: $\|\nabla_w^2 \mathcal{L}(\mathcal{S}, w)\|_F^2$

- Spectral radius: $\rho(\nabla_w^2 \mathcal{L}(\mathcal{S}, w))$

- Ours: $\widehat{\gamma}(w) = \frac{1}{T} \sum_{t=1}^{T} \ln|\xi(\mathcal{S}^t, w_0)|, \ \mathcal{S}^t \subset \mathcal{S}$

Both the Frobenius norm and the spectral radius based metric are related to ours, as from Equation 1 we have $\|\nabla_w^2 \mathcal{L}(\mathcal{S}, w)\|_F^2 = \|\mathcal{I}(w)\|_F^2$ and $\rho(\nabla_w^2 \mathcal{L}(\mathcal{S}, w)) = \rho(\mathcal{I}(w))$. These two metric, however, are too expensive to compute for the entire training set $\mathcal{S}$; we instead calculate them by averaging the results for $T$ sampled $\mathcal{S}^t \subset \mathcal{S}$, similar to when we compute $\widehat{\gamma}(w)$. We leave details of how we exactly compute these metrics in our experiments to Appendix D.

We perform evaluations in three scenarios, similar to Neyshabur et al. (2017); Keskar et al. (2017). We examine different local minima due to (1) a confusion set of varying size in training, (2) different data augmentation schemes, and (3) different batch size. In specific,

- In Scenario I, we randomly select a subset of 10000 images as the training set and train the DNN with a confusion set consisting of CIFAR-10 samples with random labels. We vary the size of the confusion set so that the resulting local minima generalize differently to the test set while all remain close-to-zero training losses. We consider confusion size of 0, 1k, 2k, 3k, 4k and 5k. We calculate all metrics based on the sampled 10000 training images.
- In Scenario II, we vary the level of data augmentation. We apply horizontal flipping, denoted `flip-only`, random cropping from images with 1 pixel padded each side plus flipping, denoted `1-crop-f`, random cropping with 4 pixels padded each side plus flipping, denoted `4-crop-f` and no data augmentation at all, denoted `no-aug`. Under all schemes, the network achieves perfect training accuracy. All the metrics are computed on the un-augmented training set.
- In Scenario III, we vary the batch size. Hoffer et al. (2017) suggests that large batch size leads to poor generalization. We consider the batch size to be 128, 256, 512 and 1024.

The default values for the 3 variables are confusion size 0, `4-crop-f` and batch size 128. For each configuration in each scenario, we train 5 models and report results (average & standard deviations) of all metrics as well as the test errors (in percentage). For the confusion set experiments, we sample a new training set and a new confusion set every time. In all scenarios, we train the model for 200 epochs with an initial learning rate 0.1, divided by 10 whenever the training loss plateaus. Within each scenario, we find the final training loss very small and very similar across different models and the training accuracy essentially equal to 1, indicating the convergence to local minima.

The results are in Figure 1, 2 and 3 for Scenario I, II and III, respectively. Our metric significantly outperforms others and is very effective in capturing the generalization properties, i.e., a lower value of our metric consistently indicates a better generalizable local minimum.

## 7.2 EXPERIMENTS ON LOCAL MINIMA REGULARIZATION

We evaluate our regularizer on CIFAR-10 & CIFAR-100 for four different network architectures including a plain CNN, ResNet-20, Wide ResNet (Zagoruyko & Komodakis, 2016) and DenseNet (Huang et al., 2017). We use WRN-28-2-B(3,3) from the Wide ResNet paper and the DenseNet-BC-k=12 from the DensetNet paper. See Appendix E for further architecture details. We denote the four networks as CNN, ResNet, WRN and DenseNet, respectively.

We manually set $\alpha = 0.0001$ in all experiments and select the other three hyper-parameters in Algorithm 1 by validation via a 45k/5k training data split for each of the network architecture on each dataset. In specific, we consider $\beta \in \{10, 20, 30, 40, 50, 75, 100, 125\}$, $M \in \{4, 8, 16\}$ and $\tau \in \{1, 5, 10, 15\}$. We keep all the other training hyper-parameters, schemes as well as the setup identical to those in their original paper (details in Appendix E). The training details of the plain CNN are also in Appendix E. We train 5 separate models for each network-dataset combination and report the test errors in percentage (mean ± std.) in Table 1, where "+reg" indicates training with our regularizer applied. The results demonstrate that our method provides consistent generalization improvement to a wide range of DNNs.

### 7.2.1 THE CHOICE OF THE OPTIMIZER

As described in Algorithm 1, our proposed regularizer is not tied to a specific optimizer. We perform experiments with SGD+Momentum because it is chosen to be used in ResNet, WRN, and DenseNet, helping all of them achieve current or previous state-of-the-art results. Our regularizer aims to find better "flatter" minima to improve generalization whereas adaptive optimization methods such as

Adam (Kingma & Ba, 2014) and AdaGrad (Duchi et al., 2011) try to boost up convergence, yet usually at the cost of generalizability. Recent works (Wilson et al., 2017; Keskar & Socher, 2017) show that adaptive methods generalize worse than SGD+Momentum. In specific, very similar to our setup, Keskar & Socher (2017) demonstrates that SGD+Momentum consistently outperforms the others on ResNet and DenseNet for CIFAR-10 and CIFAR-100. Other approaches that also utilize local curvature to improve SGD, such as the Entropy-SGD (Chaudhari et al., 2017) mentioned in Section 2, have empirical results rather preliminary compared to ours.

### 7.2.2 GENERALIZATION BOOST AS A RESULT OF BETTER LOCAL MINIMA

Our regularizer essentially optimizes an upper bound of the proposed metric during training. We perform a sanity check to illustrate that the regularizer indeed induces better local minima characterized by our metric. For ResNet, Wide-ResNet and DenseNet trained on CIFAR-10, we compute the metric on local minima of similar training loss obtained with or without applying the regularizer. Table 2 shows that the resulting generalization boost aligns with what captured by our metric.

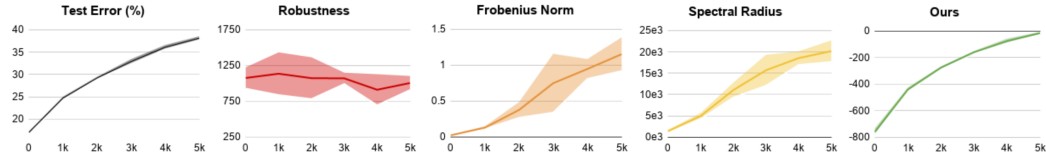

Figure 1: Scenario I: varied size of the confusion set. 5 models are trained for each size of the confusion set (x-axis). Solid lines are the average result; shaded areas represent the ± 1 standard deviation (same for Figure 2 and 3). A larger confusion set leads to a higher test error, a trend well captured by our metric and the other two; the robustness based metric fails.

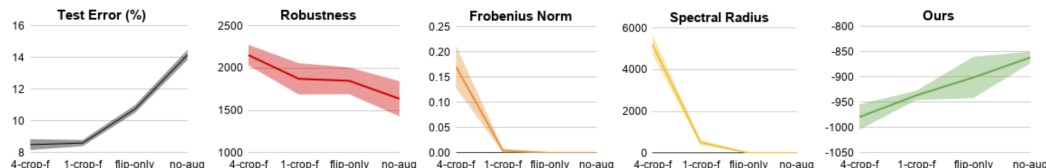

Figure 2: Scenario II: varied data augmentation schemes. Four different schemes are used. Our metric works well as an indicator of the test error while all the other metrics completely fail.

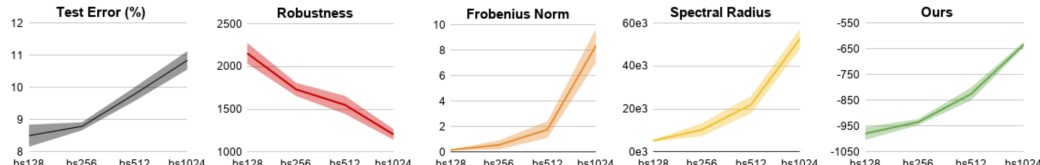

Figure 3: Scenario III: varied batch size. Large batch size leads to poor generalization, captured by all the metrics except for the robustness based one.

Table 1: Test error (%) for CIFAR-10 (1st row) and CIFAR-100 (2nd row). In general, a model with more parameters admits more space for our regularizer. The representation power of ResNet-20 is too limited for CIFAR-100 (resulting in poor convergence); so we ignore it in our experiments.

| CNN | CNN+reg | WRN | WRN+reg | DenseNet | DenseNet+reg | ResNet | ResNet+reg |
|---|---|---|---|---|---|---|---|
| $8.52 \pm 0.08$ | $\mathbf{7.46 \pm 0.13}$ | $5.44 \pm 0.04$ | $\mathbf{4.80 \pm 0.10}$ | $4.61 \pm 0.08$ | $\mathbf{4.31 \pm 0.08}$ | $8.50 \pm 0.31$ | $\mathbf{7.91 \pm 0.25}$ |
| $31.12 \pm 0.35$ | $\mathbf{29.19 \pm 0.21}$ | $25.52 \pm 0.15$ | $\mathbf{23.65 \pm 0.14}$ | $22.54 \pm 0.32$ | $\mathbf{22.19 \pm 0.28}$ | - | - |

Table 2: The proposed metric computed on local minima obained with or without applying the proposed regularizer. Each entry represents mean $\pm$ std. among 5 runs. Smaller values are bolded.

|  | ResNet | WRN | DenseNet |
|---|---|---|---|
| w/o reg. | -979.3 $\pm$ 22.3 | -737.6 $\pm$ 20.3 | -850.3 $\pm$ 23.5 |
| with reg. | **-1138.1 $\pm$ 11.0** | **-804.8 $\pm$ 18.7** | **-886.2 $\pm$ 20.5** |

## 8 CONCLUSION AND FUTURE WORK

In this paper, we show a bridge between the field of deep learning theory and regularization methods with respect to the generalizability of local minima. We propose a metric that captures the generalization properties of different local minima and provide its theoretical analysis including a generalization bound. We further derive an efficient approximation of the metric and devise a practical and effective regularizer from it. Empirical results demonstrate our success in both capturing and improving the generalizability of DNNs. Our exploration promises a direction for future work on the regularization and optimization of DNNs.

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

## A    PROOF OF EQUATION 1

To prove the second equality in Equation 1, it suffices to prove the following equality:

$$-\nabla_w^2 \ell\ell_{\mathcal{S}}(w) = \sum_{(x,y)\in\mathcal{S}} \sum_{i=1}^{K} y_i [\nabla_w \ln p(c_x = i|x; w) \nabla_w \ln p(c_x = i|x; w)^T]$$

For convenience, we change the notation of the local minimum from $w_0$ to $w$ and further denote $p(c_x = i|x; w)$ as $p_w^x(i)$. Since $-\nabla_w^2 \ell\ell_{\mathcal{S}}(w) = -\sum_{(x,y)\in\mathcal{S}} \sum_{i=1}^{K} y_i \nabla_w^2 \ln p_w^x(i)$, for each $(x, y) \in \mathcal{S}$ and $i \in \{1, 2, ..., K\}$, we have:

$$[\nabla_w^2 \ln p_w^x(i)]_{j,k} = \frac{\partial^2}{\partial w_j \partial w_k} \ln p_w^x(i)$$

$$= \frac{\partial}{\partial w_j} \left( \frac{\frac{\partial}{\partial w_k} p_w^x(i)}{p_w^x(i)} \right)$$

$$= \frac{p_w^x(i) \frac{\partial^2}{\partial w_j \partial w_k} p_w^x(i)}{p_w^x(i)^2} - \frac{\frac{\partial}{\partial w_j} p_w^x(i)}{p_w^x(i)} \frac{\frac{\partial}{\partial w_k} p_w^x(i)}{p_w^x(i)}$$

$$= \frac{\frac{\partial^2}{\partial w_j \partial w_k} p_w^x(i)}{p_w^x(i)} - \frac{\partial}{\partial w_j} \ln p_w^x(i) \cdot \frac{\partial}{\partial w_k} \ln p_w^x(i)$$

Since $w_0$ is a local minimum (also a global minimum) described in Section 4 as $y_i = p_w^x(i)$ for $i = 1, 2, ..., K$, when taking the double summation, the first term above becomes:

$$\sum_{(x,y)\in\mathcal{S}} \sum_{i=1}^{K} \frac{\partial^2}{\partial w_j \partial w_k} p_w^x(i) = \frac{\partial^2}{\partial w_j \partial w_k} \sum_{(x,y)\in\mathcal{S}} \sum_{i=1}^{K} p_w^x(i) = \frac{\partial^2}{\partial w_j \partial w_k} N = 0$$

Then it follows that:

$$[\nabla_w^2 \ell\ell_{\mathcal{S}}(w)]_{j,k} = - \sum_{(x,y)\in\mathcal{S}} \sum_{i=1}^{K} y_i [\nabla_w \ln p_w^x(i) \ \nabla_w \ln p_w^x(i)^T]_{j,k}$$

## B    PROOF OF THE GENERALIZATION BOUND IN SECTION 5.2

First let us review the PAC-Bayes Theorem in McAllester (2003):

**Theorem 2.** *For any data distribution $\mathcal{D}$ and a loss function $\mathcal{L}(\cdot, \cdot) \in [0, 1]$, let $\mathcal{L}(\mathcal{D}, w)$ and $\mathcal{L}(\mathcal{S}, w)$ be the expected loss and training loss respectively for the model paramterized by $w$, with the training set $|\mathcal{S}| = N$. For any prior distribution $\mathcal{P}$ with a model class $\mathcal{C}$ as its support, any posterior distribution $\mathcal{Q}$ over $\mathcal{C}$ (not necessarily Bayesian posterior), and for any $\delta \in (0, 1]$, we have with probability at least $1 - \delta$ that:*

$$\mathop{\mathbb{E}}_{w\sim\mathcal{Q}}[\mathcal{L}(\mathcal{D}, w)] \leq \mathop{\mathbb{E}}_{w\sim\mathcal{Q}}[\mathcal{L}(\mathcal{S}, w)] + 2\sqrt{\frac{2D_{\mathrm{KL}}(\mathcal{Q}||\mathcal{P}) + \ln\frac{2N}{\delta}}{N-1}}$$

As $e^{\gamma(w_0)} = |\mathcal{I}(w_0)|$, we can rewrite the generalization bound we want to prove in Section 5.2 as:

$$
\mathbb{E}_{w \sim \mathcal{Q}}[\mathcal{L}(\mathcal{D}, w)] \le \mathcal{L}_0 + \frac{W \cdot V^{2/W} \pi^{1/W} |\mathcal{I}(w_0)|^{1/W}}{4\pi e}
$$
$$
+ 2\sqrt{\frac{W \cdot V^{2/W} \pi^{1/W} |\mathcal{I}(w_0)|^{1/W} + 4\pi e \mathcal{L}_0 + 2\pi e \ln \frac{2N}{\delta}}{2\pi e (N-1)}}
\tag{7}
$$

As defined in Section 5.2, given the model class $\mathcal{M}(w_0)$, whose volume is $V$, for the neural network $f_w$, the uniform prior $\mathcal{P}$ attains the probability density function $p(w) = \frac{1}{V}$ for any $w \in \mathcal{M}(w_0)$ and the posterior $\mathcal{Q}$ has density $q(w) \propto e^{-|\mathcal{L}(\mathcal{S}, w) - \mathcal{L}_0|}$. Based on Assumption 2 and the observed Fisher information $\mathcal{I}(w_0)$ derived in Section 4, especially the Equation 2, we have:

$$
\mathcal{L}(\mathcal{S}, w) = \mathcal{L}_0 + \frac{1}{2}(w - w_0)^T \mathcal{I}(w_0)(w - w_0) \quad \forall w \in \mathcal{M}(w_0)
$$

Denote $\Sigma = [\mathcal{I}(w_0)]^{-1} = [\nabla_w^2 \mathcal{L}(\mathcal{S}, w_0)]^{-1}$. Then $\mathcal{Q}$ is a truncated multivariate Gaussian distribution whose density function $q$ is:

$$
q(w; w_0, \Sigma) = \frac{\sqrt{(2\pi)^{-n} |\Sigma|^{-1}} \exp\{-\frac{1}{2}(w - w_0)^T \Sigma^{-1}(w - w_0)\}}{\int_{\mathcal{M}(w_0)} \sqrt{(2\pi)^{-n} |\Sigma|^{-1}} \exp\{-\frac{1}{2}(w - w_0)^T \Sigma^{-1}(w - w_0)\} \, dw}
$$
$$
= \frac{\exp\{-\frac{1}{2}(w - w_0)^T \Sigma^{-1}(w - w_0)\}}{\int_{\mathcal{M}(w_0)} \exp\{-\frac{1}{2}(w - w_0)^T \Sigma^{-1}(w - w_0)\} \, dw}
\tag{8}
$$

Denote the denominator of Equation 8 as $\mathbf{Z}$ and define:

$$
g(w; w_0, \Sigma) \triangleq -\frac{1}{2}(w - w_0)^T \Sigma^{-1}(w - w_0)\} \le 0
$$

Then $q$ can also be written as:

$$
q(w; w_0, \Sigma) = \frac{\exp\{g(w; w_0, \Sigma)\}}{\mathbf{Z}}
$$

In order to derive a generalization bound in the form of the PAC-Bayes Theorem, it suffices to prove an upper bound of the KL divergence term:

$$
\begin{aligned}
D_{\mathrm{KL}}(\mathcal{Q}||\mathcal{P}) &= \mathbb{E}_{w \sim \mathcal{Q}} \ln \frac{q(w)}{p(w)} \\
&= -\mathbb{E}_{w \sim \mathcal{Q}} \ln \frac{1}{V} + \mathbb{E}_{w \sim \mathcal{Q}} \ln q(w) \\
&= \ln V + \mathbb{E}_{w \sim \mathcal{Q}} g(w; w_0, \Sigma) + \ln \frac{1}{\mathbf{Z}} \\
&\le \ln V + \mathbb{E}_{w \sim \mathcal{Q}} 0 - \ln \left( \int_{\mathcal{M}(w_0)} \exp\{g(w; w_0, \Sigma)\} \, dw \right) \\
&\le \ln V - \ln \left( \int_{\mathcal{M}(w_0)} \exp\{-\max_{w \in \mathcal{M}(w_0)} \mathcal{L}(\mathcal{S}, w)\} \, dw \right) \\
&= \ln V - \ln \left( V \cdot \exp\{-\max_{w \in \mathcal{M}(w_0)} \mathcal{L}(\mathcal{S}, w)\} \right) \\
&= \ln V - \ln V + h \quad = \quad h
\end{aligned}
$$

where $h$ is the height of $\mathcal{M}(w_0)$ defined in Section 5.1. For convenience, we shift down $\mathcal{L}(\mathcal{S}, w)$ by $\mathcal{L}_0$ and denote the shifted training loss $\mathcal{L}_0(w) \triangleq \mathcal{L}(\mathcal{S}, w) - \mathcal{L}_0$ so that $\mathcal{L}_0(w_0) = 0$. Then

$$
\mathcal{L}_0(w) = \frac{1}{2}(w - w_0)^T \Sigma^{-1}(w - w_0) \quad \forall w \in \mathcal{M}(w_0)
$$

Furthermore, the following two sets are equivalent

$$
\{w \in \mathbb{R}^W : \mathcal{L}(\mathcal{S}, w) = h\} = \{w \in \mathbb{R}^W : \mathcal{L}_0(w) = h - \mathcal{L}_0\}
$$

both of which are the $W$-dimensional hyperellipsoid given by the equation $\mathcal{L}_0(w) = h - \mathcal{L}_0$, which can be converted to the standard form for hyperellipsoids as:

$$(w - w_0)^T \frac{\Sigma^{-1}}{2(h - \mathcal{L}_0)} (w - w_0) = 1$$

The volume enclosed by this hyperellipsoid is exactly the volume of $\mathcal{M}(w_0)$, i.e., $V$; so we have

$$\frac{\pi^{W/2}}{\Gamma(\frac{W}{2} + 1)} \sqrt{2^W (h - \mathcal{L}_0)^W |\Sigma|} = V$$

Solve for $h$, with the Stirling's approximation for factorial $\Gamma(n+1) \approx \sqrt{2\pi n} \left(\frac{n}{e}\right)^n$, we have

$$h = \mathcal{L}_0 + \frac{(V \cdot \Gamma(\frac{W}{2} + 1))^{2/W}}{2\pi |\Sigma|^{1/W}} \approx \mathcal{L}_0 + \frac{V^{2/W} \pi^{1/W} W^{(W+1)/W} |\mathcal{I}(w_0)|^{1/W}}{4\pi e}$$

where $\Gamma(\cdot)$ denotes the Gamma function. Notice that for modern DNNs we have $W \gg 1$, and so $W^{\frac{W+1}{W}} \approx W$. Then the generalization bound in the form of the PAC-Bayes Theorem is given as:

$$\mathbb{E}_{w \sim \mathcal{Q}}[\mathcal{L}(\mathcal{D}, w)] \leq \mathbb{E}_{w \sim \mathcal{Q}}[\mathcal{L}(\mathcal{S}, w)] + 2\sqrt{\frac{W \cdot V^{2/W} \pi^{1/W} |\mathcal{I}(w_0)|^{1/W} + 4\pi e \mathcal{L}_0 + 2\pi e \ln \frac{2N}{\delta}}{2\pi e (N-1)}}$$

We can further bound the first term on the right hand side as:

$$\mathbb{E}_{w \sim \mathcal{Q}}[\mathcal{L}(\mathcal{S}, w)] \leq \mathbb{E}_{w \sim \mathcal{Q}}[\max_{w \in \mathcal{M}(w_0)} \mathcal{L}(\mathcal{S}, w)] = h$$

Putting it all together, we can finally obtain Equation 7.

## C   DERIVATION OF EQUATION 5

First, let us present the well-known theorem in linear algebra that relates the eigenvalues of a matrix to those of its sub-matrices.

**Theorem 3.** *Given an $n \times n$ real symmetric matrix $A$ with eigenvalues $\lambda_1 \leq ... \leq \lambda_n$, for any $k < n$ denote its principal sub-matrix as $B$ obtained from removing $n - k$ rows and columns from $A$. Let $\nu_1 \leq ... \leq \nu_k$ be the eigenvalues of $B$. Then for any $1 \leq r \leq k$, we have $\lambda_r \leq \nu_r \leq \lambda_{r+n-k}$.*

Let $\{\nu_n\}_{n=1}^{N'}$ be the eigenvalues of $\frac{1}{W} \xi^t(w_0)$, which is a $N' \times N'$ sub-matrix of $\mathcal{I}'(w_0)$; then

$$\widehat{\gamma}(w_0) = \frac{1}{T} \sum_{t=1}^{T} \ln |\xi^t(w_0)| = \frac{1}{T} \sum_{t=1}^{T} \ln |W \cdot \frac{1}{W} \xi^t(w_0)| = N' \ln W + \frac{1}{T} \sum_{t=1}^{T} \sum_{n=1}^{N'} \ln \nu_n$$

Theorem 2 gives the relation between $\nu_n$ and $\lambda_n$, defined above and in Section 5.3 as the $n^{\text{th}}$ smallest eigenvalues of $\frac{1}{W} \xi^t(w_0)$ and that of $\mathcal{I}'(w_0)$, respectively. For sufficiently large $N'$, we can use $\nu_n$ to approximate $\lambda_n$. Ignoring the eigenvalues of $\mathcal{I}'(w_0)$ larger than $\lambda_{N'}$ is reasonable when estimating $\gamma(w_0)$, since in general the majority of the eigenvalues of the Hessian for DNNs are close to zero with only a few large "outliers" (Pennington & Worah, 2018; Sagun et al., 2018), and so the smallest eigenvalues are the dominant terms in $\gamma(w_0)$. A specific bound of the eigenvalues remains an open question, though. In short, we have $\sum_{n=1}^{N'} \nu_n \approx \sum_{n=1}^{N'} \lambda'_n$ and consequently:

$$\frac{W}{N'} \widehat{\gamma}(w_0) + W \ln \frac{1}{W} = \frac{W}{N'} \widehat{\gamma}(w_0) - W \ln W$$

$$= \frac{W}{N'} \left( \widehat{\gamma}(w_0) - N' \ln W \right)$$

$$= \frac{1}{T} \sum_{t=1}^{T} \frac{W}{N'} \sum_{n=1}^{N'} \ln \nu_n$$

$$\approx \frac{1}{T} \sum_{t=1}^{T} \frac{W}{N'} \sum_{n=1}^{N'} \ln \lambda'_n$$

Finally we we have

$$\lim_{T \to \infty} \frac{1}{T} \sum_{t=1}^{T} \frac{W}{N'} \sum_{n=1}^{N'} \ln \lambda'_n = \gamma(w_0)$$

## D    DETAILS OF CALCULATING THE METRICS IN SECTION 7.1

For the following metrics, we apply estimation by sampling a subset $\mathcal{S}^t$ from the full training set $\mathcal{S}$ for $T$ times and averaging the results:

- Frobenius norm: $\|\nabla^2_w \mathcal{L}(\mathcal{S}, w)\|^2_F$
- Spectral radius: $\rho(\nabla^2_w \mathcal{L}(\mathcal{S}, w))$
- Ours: $\widehat{\gamma}(w) = \frac{1}{T} \sum_{t=1}^{T} \ln|\xi(\mathcal{S}^t, w_0)|$

For the Frobenius norm based metric, from Equation 2 we have:

$$\|\nabla^2_w \mathcal{L}(\mathcal{S}, w)\|^2_F = \|\mathcal{I}(w)\|^2_F = \frac{1}{N} \sum_{(x,y)\in\mathcal{S}} \sum_{i=1}^{K} \left\| (\nabla_w[\boldsymbol{\ell}_x(w_0)]_i)(\nabla_w[\boldsymbol{\ell}_x(w_0)]_i)^T \right\|^2_F$$

Similar to Equation 4, we approximate $y$ by $\tilde{y}$ and so

$$\|\nabla^2_w \mathcal{L}(\mathcal{S}, w)\|^2_F \approx \frac{1}{N} \sum_{(x,y)\in\mathcal{S}} \left\| (\nabla_w[\boldsymbol{\ell}_x(w_0)]_{\bar{y}})(\nabla_w[\boldsymbol{\ell}_x(w_0)]_{\bar{y}})^T \right\|^2_F$$

Summing over the entire Hessian matrix is too expensive as there are $W \times W \times N$ entries in total. We therefore estimate the quantity by first sampling a subset $\mathcal{S}^t \subset \mathcal{S}$ and then sampling 100,000 entries of $(\nabla_w[\boldsymbol{\ell}_x(w_0)]_{\bar{y}})(\nabla_w[\boldsymbol{\ell}_x(w_0)]_{\bar{y}})^T$. We perform the estimation $T$ times and average the results, similar to the approach when computing $\widehat{\gamma}(w)$.

Also by Equation 2 and the approximation in Equation 4, the spectral radius of Hessian is equivalent to the squared spectral norm of $1/\sqrt{N}\mathbf{J}_w[\tilde{\mathcal{L}}(\mathcal{S}, w)]$. We also perform estimation (with irrelevant scaling constants dropped) by sampling $\mathcal{S}^t$ for $T$ times, i.e., via $\frac{1}{T}\sum_t \|\mathbf{J}_w[\tilde{\mathcal{L}}(\mathcal{S}^t, w)]\|^2_2$.

Furthermore, in all our experiments that involves samplings $\mathcal{S}^t$, we set $|\mathcal{S}^t| = N' = T = 100$.

## E    ARCHITECTURE AND TRAINING DETAILS IN SECTION 7

Architecture details are as below

- The plain CNN is a 6-layer convolutional neural network similar to the baseline in Lee et al. (2016) yet without the "mlpconv" layers (resulting in a much fewer number of parameters). Specifically, the 6 layers has numbers of filters as $\{64, 64, 128, 128, 192, 192\}$. We use $3 \times 3$ kernel size and ReLU as the activation function. After the second and the fourth convolutional layer we insert a $2 \times 2$ max pooling operation. After the last convolutional layer, we apply a global average pooling before the final softmax classifier.
- For ResNet-20 and WRN-28-2-B(3,3), we use the same architecture as in their original papers, with the only difference that we use pre-activation as in He et al. (2016b). This results in slightly stronger baselines than the models in their original papers.
- For DenseNet-BC-k=12 we use the the architecture identical to the one used in the original paper.

The training details are

- For the plain CNN, we initialize the weights according to the scheme in He et al. (2016a) and apply l2 regularization of a coefficient 0.0001. We perform standard data augmentation, the one denoted `4-crop-f` in Section 7.1. We use stochastic gradient descent with Nesterov momentum set to 0.9 and a batch size of 128. We train 200 epochs in total with the learning rate initially set to 0.01 and then divided by 10 at epoch 100 and 150.
- For ResNet-20, WRN-28-2-B(3,3) and DenseNet-BC-k=12, we use the same hyperparameters, training schemes, data augmentation schemes, optimization methods, etc., as those in their original papers, respectively.

