# OpenReview forum: "Information-Theoretic Local Minima Characterization and Regularization"
_ICLR.cc/2020/Conference — Reject_

### Official Review · AnonReviewer1 · 2019-10-21
**Official Blind Review #1**

**Rating:** 8

**Review:**

This paper contributes to the deep learning generalization theory, mainly from the theoretical perspective with experimental verifications. The key proposition is given by the unnumbered simple equation in the middle of page 4 (please number it), where \mathcal{I} is the Fisher information matrix. According to the authors, this simple metric, which is the log-determinant of the Fisher information matrix, can characterize the generalization of a DNN.

Remarkably, this piece of work is well written in terms of English and formulations, and complete, with a rigorous theoretical analysis (section 5.1, 5.2), practical approximations (section 5.3) and empirical verifications (section 6).

On the theoretical side, this work builds upon Rissanen's formulation of the MDL principle, which has two parts (describing data given the model as well as the model complexity). Under rough approximations, the complexity term becomes the log-determinant of the Fisher information matrix evaluated at the local (global) optimum. This simple approximation is further proved to upper-bounds the generalization error as stated in theorem 1.

To make the criterion to be practically useful, the author used the Jensen inequality so that the metric simply depends on the trace of the Fisher information matrix.

The empirical study showed the usefulness of the proposed metric which can well approximate the testing error and a regularization term (based on the trace of the Fisher information matrix) that can improve generalization on real DNN experiments.

The reviewer has the following minor comments to further improve this contribution:

section 5.1, explain the abbreviation FIA

Regarding the choice of the neighborhood \mathcal{M}(w_0), what is the reason to define the model (neighbourhood of w_0) based on the loss? Why not simply take a coordinate neighborhood?

According to your metric, the smaller the scale of the Fisher information matrix, the better the generalization. In section 5.1, there has to be some remarks on the intuition and related works on the flatness of the local minimum that is related to generalization.

As this contribution is related to the spectral properties of the Fisher information matrix, the reviewer points the authors to "Universal Statistics of Fisher Information in Deep Neural Networks: Mean Field Approach. Karakida et al. 2018." and "Lightlike Neuromanifolds, Occam's Razor and Deep Learning. Sun and Nielsen. 2019", which deals with asymptotic cases and have similar MDL formulations expressed in terms of the spectrum of the Fisher information matrix.


**Experience Assessment:**

I have published one or two papers in this area.

**Review Assessment: Checking Correctness Of Derivations And Theory:**

I assessed the sensibility of the derivations and theory.

**Review Assessment: Checking Correctness Of Experiments:**

I assessed the sensibility of the experiments.

**Review Assessment: Thoroughness In Paper Reading:**

I read the paper at least twice and used my best judgement in assessing the paper.

---

> ### Author Response · Authors · 2019-11-12
> **Response to reviewer #1**
>
> Dear reviewer1,
>
> Thanks for your appreciation.
>
> Q1: “Section 5.1, explain the abbreviation FIA”
>
> FIA in Section 5.1 stands for Fisher information approximation, originally coined for Normalized Maximum Likelihood Estimation in [1].
>
> Q2: “Regarding the choice of the neighborhood \mathcal{M}(w_0), what is the reason to define the model (neighbourhood of w_0) based on the loss? Why not simply take a coordinate neighborhood?”
>
> Given the local minimum at w_0, we find it natural to define its neighborhood by a sublevel set w.r.t. the training loss. The issue of using the local coordinate (e.g., using an Ɛ-ball to define the neighborhood) is that the amount of change of the underlying model measured by training loss varies for different dimensions of the parameter space. For instance, moving in one direction might change the model a lot while moving in the other might change little.
>
> Q3: “In section 5.1, there have to be some remarks on the intuition and related works on the flatness of the local minimum that is related to generalization.”
>
> We will update the paper to add a discussion in Section 5.1. regarding the intuition and related works on “flatness/sharpness”, some of which are briefly discussed in Section 2.
>
> Q4: “the reviewer points the authors to [2] and [3], which have similar MDL formulations expressed in terms of the spectrum of the Fisher information matrix”
>
> We will definitely consider mentioning the relation with these two papers in our next version.
>
> [1] Rissanen, Jorma J. "Fisher information and stochastic complexity." IEEE transactions on information theory 42.1 (1996): 40-47.
>
> [2] Karakida, Ryo, Shotaro Akaho, and Shun-ichi Amari. "Universal Statistics of Fisher Information in Deep Neural Networks: Mean Field Approach." The 22nd International Conference on Artificial Intelligence and Statistics. 2019.
>
> [3] Sun, Ke, and Frank Nielsen. "Lightlike Neuromanifolds, Occam's Razor and Deep Learning." arXiv preprint arXiv:1905.11027 (2019).

---

> > ### Comment · AnonReviewer1 · 2019-11-15
> > **Acknowlging rebuttal and a few remarks**
> >
> > Thank you for the revision, which has addressed most of my comments and substantially enhanced the presentation.
> >
> > I am writing here some further remarks here for the next round of revision or for potential future works.
> >
> > For the definition of the observed Fisher information, the authors may consider cite " Efron & Hinkley. Assessing the accuracy of the maximum likelihood estimator: Observed versus expected Fisher Information. 1978".
> >
> > Somewhere around 5.1.1 (in the revised draft), it has to be mentioned the volume element defined by the Fisher information (including the observed information) that is \sqrt{\vert{I}(\theta)\vert}d\theta, and the total volume, the exponential of the last term of the FIA, is invariant to reparameterization. In the FIA, w_0 should be the maximum likelihood estimation (that means the global optimum).
> >
> > In the writing, it may help to emphasize that 5.1.1 is only a rough approximation to show the relationship between the proposed \gamma quantity and the FIA, as the approximation is rough and the quality is not guaranteed, and the theoretical guarantee is given by 5.2.  Actually, the authors may also consider approximate \gamma based on Balasubramanian's book chapter "MDL, Bayesian Inference and the Geometry of the Space of Probability Distributions", where there is an explicit term of the log determinant of the observed Fisher information.
> >
> > Regarding the bound in section 6. It is better to have some remarks and/or numerical simulations (e.g. in the appendix) on the tightness of the bound so that the reader can have some intuitions. A potential extension is a variational bound (with free parameters).
> >
> > Overall, I don't think this paper in its current form has any major flaws and the contribution is valid with both a theorem and empirical results. It should be interesting to the ICLR community and could enlighten discussions.

---

### Official Review · AnonReviewer2 · 2019-10-22
**Official Blind Review #2**

**Rating:** 1

**Review:**

Post-rebuttal update: I have just noticed the authors modified their summary post below and claimed "[my concerns] are all minor or resolved". This is not true. Here is my summary of unresolved concerns written after the discussion period.

This work has been substantially improved during the rebuttal process, and some of my concerns are addressed. But there are still major issues, as raised in my [last comment]( https://openreview.net/forum?id=BJlXgkHYvS&noteId=r1xAnokijS ), that remains unanswered. Specifically,

(A) the relation between this work and information theory

In the revision, the authors have make it very clear that the relation between FIA and their proposed regularized objective is very vague, relying on the crude approximation of expected Fisher information with observed Fisher information. Therefore the "information-theoretic" part in the title seems awkward and to some extent, misleading.

As Reviewer 1 has pointed out, it would have been better if the authors relate their theory and method to the observed FIM, instead of information theory, from the beginning. Since the observed FIM and the neural tangent kernel (NTK) share the same eigenspectrum, it would also be interesting to relate this work to the NTK.

(B) the different behavior of the proposed regularization (log det(I)) and its bound that is actually implemented (log tr(I))

This is the more important issue. My concern is that the observed FIM (or the NTK) is known to have fast decaying spectrum; (Karakida et al) has shown empirically that the decay can be exponential. Thus log det(I) would be dominated by the long tail (since after taking logarithm it is the sum (or average) of an arithmetic sequence), while log tr(I) would be dominated by the first largest few values.

The authors claim that this is not an issue since they replaced the observed FIM with a subsampled, low-rank (<=10), version. It corresponds to consider a small submatrix of (the gram matrix of) the NTK. Denote this matrix as .
(a) This does not help with the problem, since we now have no chance of recovering the smaller eigenvalues that would have dominated log(det(I)), and it is impossible that the proposed regularizer has a similar behavior to log(det(I)).
(b) One could verify easily, using small feed-forward networks (or even simpler, computing a gram matrix using RBF kernels, since the FIM shares its eigenspectrum with NTK which is a p.d. kernel), that the new matrix  still has a fast-decaying eigenspectrum, so the behavior of  and \tilde{I} are still significantly different, even though this cannot be established by concentration bounds as the authors argue. While FFN and modern deep architectures can have different behaviors, I believe the above evidence suggests that a numerical experiment comparing the behavior of the two bounds is a must.

Following this argument we can see *another issue* of this work, namely the proposed generalization bound will be vacuous given the fast-decaying spectrum of the FIM, since it contains gamma=log(det(I)).

Reviewer 1 mentioned this work could enlighten future discussions on this subject. While I agree this paper presents interesting empirical observations (namely its final algorithm, which is vaguely connected to the proposed objective, leads to improved performance on CV tasks), I think this submission in its current form is a bit too misleading to serve this purpose well, and overall I believe it would be better to go through another round of revision.


Original Review
============================================

This paper presents a generalization bound based on Fisher information at a local optima, and proposes to optimize (an approximation to) it to get better generalization guarantees. There are issues in both parts, and I don't think it should be accepted. Specifically,

1. The definition of Fisher information is incorrect (for almost every parameter). The expectation should be taken w.r.t the model distribution p(c_x|x;w), instead of the data distribution S.
2. Assumption (1) (loss locally quadratic) is not reasonable for DNNs, since local optimas will not be unique in their neighborhoods. See e.g. Section 12.2.2, "Information Geometry and Its Applications".
3. Regarding the approximation to the bound, approximating log det(I) with log trace(I) is not a good idea: adding a very small eigenvalue will lead to noticeable change in the former, but negligible change in the latter. This is particularly problematic for DNNs, since the spectrum of their Fisher information matrix varies in a wide range: see "Universal Statistics of Fisher Information in Deep Neural Networks: Mean Field Approach".

(Edit 11.8:
* regarding point (1), there is a quantity called observed Fisher information in e.g. Grunwald (2007) that coincide with Eq (1) in the paper, but it is a function of the dataset instead of the model parameter, and can only used to study model parameters near the *global optima* (as it is applied in Grunwald (2007)); it cannot help with choosing between different local optimas as this work claims. Additionally, the FIA criterion, which is used in this paper to devleop the generalization bound, is defined using the standard form of Fisher information (i.e. taking expectation w.r.t model distribution), see Rissanen (1996). These facts lead me to believe this is a confusion on the authors' part.
* in point (2) I was referring to the authors' argument " Since L(S,w) is analytic and w_0 is the *only local minimum* of L(S,w) in M(w_0)", which is incorrect.)

**Experience Assessment:**

I have read many papers in this area.

**Review Assessment: Checking Correctness Of Derivations And Theory:**

I carefully checked the derivations and theory.

**Review Assessment: Checking Correctness Of Experiments:**

I did not assess the experiments.

**Review Assessment: Thoroughness In Paper Reading:**

I read the paper at least twice and used my best judgement in assessing the paper.

---

> ### Author Response · Authors · 2019-11-12
> **Response to reviewer #2**
>
> Dear reviewer2,
>
> Thanks for your time and effort. There seems to be quite a lot of misunderstandings and confusions from the reviewer.
>
> Q1: “The definition of Fisher information is incorrect … The expectation should be taken w.r.t the model distribution”.
>
> The statement in this question is factually wrong. First of all, what we use in the paper is the observed Fisher information, not the (expected) Fisher information. Secondly, by definition, the Fisher information, no matter the observed or the expected one, has nothing to do with expectation w.r.t. the model distribution.
>
> Q2: “There is a quantity called observed Fisher information that coincide with Eq (1) in the paper.”
>
> This is by no means a coincidence. We clearly and precisely describe the term as observed Fisher information in the very first place when we introduce Eq (1).
>
> Q3: “The observed Fisher information is a function of the dataset instead of the model parameter, as in Grünwald (2007)”
>
> This is factually wrong. Professor Peter Grünwald has never said so. The Fisher information, no matter the observed one or the expected version, is a quantity involving both the model parameters and the input data. In Grünwald (2007), simply omitting model parameter θ in the notation I(X) does not mean I(X) is not a function of θ.
>
> Q4: “The observed Fisher information can only used to study model parameters near the global optima; it cannot help with choosing between different local optima as the work claims”
>
> It is clearly stated in Section 4 that the different local minima we focus on to compare are also global minima. The comparison between these local minima is well-motivated, as pointed out at the beginning of Section 1, that learning algorithms such as SGD tend to end up in one of the many local (global) minima that are not distinguishable from their similar close-to-zero training loss [2, 3, 4, 5].
>
> Q5: “The FIA criterion, which is used in this paper to develop the generalization bound, is defined using the expected Fisher information rather than the observed one.”
>
> We give the FIA criterion precisely in Section 5.1, only to illustrate the connection between our approach and Rissanen's formulation of the MDL principle. In fact, we do not use the FIA criterion to derive or describe the generalization bound.
>
> Q6: “local optima will not be unique in their neighborhoods, as in [1]"
>
> In our paper, we assume that local minima we care about are well isolated, mentioned at the end of Section 5.1. For state-of-the-art network architectures used in practice, this isolation assumption is often the fact. The reviewer pointed out that, introduced in [1], two kinds of singularity in neural networks prevent the local minima from being unique, namely the eliminating singularity and the overlapping singularity. As well demonstrated in [6], network with skip connections (such as ResNet, WRN, and DenseNet used in our experiments) can effectively eliminate both. We would like to add a discussion paragraph about the isolation assumption in our paper.
>
> Q7: “Regarding the approximation to the bound, approximating log det(I) with log trace(I) is not a good idea.”
>
> We do not use log trace(I) as an approximation to measure local minima as indeed it can be inaccurate. Instead, we use it as an upper bound of what we intend to optimize during training. Optimizing such upper bound, in return, enables us to develop a tractable regularization technique in search of the good local minima. Our experiments in Section 7.2 well demonstrate the effectiveness of our proposed regularizer in finding better local minima of greater generalizability.
>
>
> [1] Amari, Shun-ichi. Information geometry and its applications. Vol. 194. Berlin: Springer, 2016.
>
> [2] Dauphin, Yann N., et al. "Identifying and attacking the saddle point problem in high-dimensional non-convex optimization." Advances in neural information processing systems. 2014.
>
> [3] Kawaguchi, Kenji. "Deep learning without poor local minima." Advances in neural information processing systems. 2016.
>
> [4] Nguyen, Quynh, and Matthias Hein. "Optimization Landscape and Expressivity of Deep CNNs." International Conference on Machine Learning. 2018.
>
> [5] Du, Simon S., et al. "Gradient Descent Finds Global Minima of Deep Neural Networks." International Conference on Machine Learning, 2019.
>
> [6] Orhan, A. Emin, and Xaq Pitkow. "Skip connections eliminate singularities." International Conference on Learning Representations, 2018.

---

> > ### Comment · AnonReviewer2 · 2019-11-12
> > **Quick Response regarding FIM Definition; Complete Response will Follow**
> >
> > Thank you for your response. I will upload a detailed response in 1-2 days regarding all aspects in your rebuttal, but I'd like to point out your response regarding the definition of Fisher information is simply incorrect.
> >
> > > the Fisher information, no matter the observed *or the expected one*, has nothing to do with expectation w.r.t. the model distribution
> >
> > See e.g.
> > * (Karakida et al, 2018), below Eqn (1):
> > "The expectation E[·] is taken over the input-output pairs (x, y) of the joint distribution p(x, y; θ)."
> > * (Grünwald, 2007), Eqn (4.18) (regarding the expected FIM), where there is a subscript of theta in the expectation; this is further clarified in Eq 18.48 (expected FIM, equivalent to Eq 4.18 assuming suitable differentiability), where you can see the model density appears in the integral.
> > * Eq 10.10 and 10.11, "All of Statistics: a Concise Course in Statistical Inference", electronic version from https://www.ic.unicamp.br/~wainer/cursos/1s2013/ml/livro.pdf , where, again, the model density appears in the integral.
> >
> > While it may be valid to formally use the observed FIM as a model selection criteria, it will lose the information theoretic interpretation as I've argued, and would be hugely misleading in my opinion. I would also need to re-check the proof of your results.
> >
> > References:
> >
> > Karakida et al: Universal Statistics of Fisher Information in Deep Neural Networks: Mean Field Approach

---

> > > ### Author Response · Authors · 2019-11-12
> > > **The reviewer #2 might be confused about some basic concepts**
> > >
> > > Thanks for the quick reply. After reading the response, we believe the reviewer might have notions of basic concepts different from those of most researchers in this field. There are a lot of textbooks out there as references. For the convenience of the reviewer, we will use the one he/she provided, referred to as [7].
> > >
> > > Q: “... model distribution ...”
> > >
> > > The reviewer here confuses the concept of probability mass (density) function of the data with that of the model. p(x; θ) is called the probability mass function (of the data), not the “model distribution”; similarly, f(x; θ) is called the probability density function (of the data), not the “model density”. The data distribution refers to the distribution of the data; similarly, the model distribution refers to the distribution of the model. The model here refers to a statistical model, which, as given in Section 7.2 of [7], is a set of parameters Θ or a set of densities. Each density here refers to a specific probability density function of the data, denoted f(x; θ) and attained by specific model parameters θ in the parameter set Θ. The model distribution here can refer to either the prior p(θ) or the posterior p(θ|x).
> > >
> > > Q2: “[our] response regarding the definition of Fisher information is simply incorrect.”
> > >
> > > According to the previous answer, our response is correct.
> > >
> > > Q3: “Using the observed FIM will lose the information-theoretic interpretation… and would be hugely misleading.”
> > >
> > > We wonder what exactly would be misleading. The proposed generalization bound does not require any information-theoretic interpretation to be correct. It is derived solely based on the theory of PAC-Bayes.
> > >
> > > [7] "All of Statistics: a Concise Course in Statistical Inference", electronic version from https://www.ic.unicamp.br/~wainer/cursos/1s2013/ml/livro.pdf

---

> > > > ### Comment · AnonReviewer2 · 2019-11-12
> > > > **I see there is a confusion about notion, but my points still hold and please revise Sec 5.1**
> > > >
> > > > Indeed by model distribution I meant the conditional data distribution p(y|x;\theta)p(x), not any distribution over the model parameters, which we don't have any to begin with. Still, I'm not completely certain this is a "nonstandard notation" - a Google search produces the following paper using the same notion: arxiv:1905.12558, arxiv:1906.07774; and in any case, I've made it clear from my first comment that I was referring to this distribution ("the model distribution p(c_x|x;w)", which, to be fully clear, refers to the joint distribution p(c_x|x;w)p(x)).
> > > >
> > > > Also, note that in Section 5.1 you used the expected Fisher information and referred to it using the same notation, which is confusing, irrelavent to the rest of the paper, and cannot be used to justify the work. You also stated that you are approximating it in the end of Section 5.1. By "lose the information theoretic interpretation" I was also referring to this part, since this is the only place where "information theory" appeared in the entire paper. In this regard, I believe the "information theoretic" part in the title as well as this section should be revised / removed to clarify this.

---

> > ### Comment · AnonReviewer2 · 2019-11-12
> > **Another quick response; full response will follow later**
> >
> > Q7: “Regarding the approximation to the bound, approximating log det(I) with log trace(I) is not a good idea.”
> >
> > I was referring to the process of *bounding* log det(I) with log trace(I), after appropriate re-scalings, since following my argument, it may not lead to informative gradients. Of course this is a upper bound following Jensen's inequality, but this mere fact would not be a sufficient justification - otherwise we would be training variational autoencoders without using encoders at all, instead relying on the variational bound using the prior p(z). The fact that your proposed regularization works could very possibly be attributed to other factors; it might even be superior to the true regularizer (log det(I)) since they have such different behaviors, with log det(I) focusing on the average eigenvalue and log tr(I) focusing on the a few largest ones. This is particularly problematic, since, as I've mentioned in the original comment, the eigenvalues of the FIM varies in a wide range.
> >
> > We will need a link between them since your theory is about the original one. If a convincing argument is to be made, I would recommend to look at toy networks where calculating log det(I) is possible, as well as toy, potentially 1D, datasets, and compare the behavior of the two regularizers.

---

> ### Author Response · Authors · 2019-11-13
> **A final summary of the review #2 and the corresponding responses from the authors**
>
> The reviewer #2 has raised three concerns, of which all are minor and resolved (completely or partially).
>
> CONCERN (I): the Fisher information is given incorrectly in the paper
>
> Both the reviewer and the authors now agree that this concern is due to the initial confusion of the reviewer.  The Fisher information used in the paper is given precisely and correctly. Most previous discussions between the reviewer and the authors start because of the different terminologies being used. The reviewer argues that their definition of “model distribution” is not “nonstandard” by pointing out two papers on ArXiv, of which one considers the model distribution as p(x, y; θ) and the other defines it as p(y | x; θ), already an ambiguity.
>
> CONCERN (II): the relation and role of Information Theory in the lens of the FIA criterion in our paper
>
> We have updated the Section 5.1 according to the request to clarify the relation of the FIA criterion and our propose metric. We would like to reiterate that the FIA criterion is NOT used in the paper to derive any theory or practice. The entire paper does NOT require any information-theoretic interpretation to be correct. The FIA criterion is stated as an important related work of our approach.
>
> CONCERN (III): the proposed regularizer that optimizes the upper bound tr(I), although works great in practice, might have underlying behavior quite different from directly optimizing det(I), which is intractable though.
>
> The reviewer refers to [1] for the spectral density of the observed FIM, denoted I, and [2] for the following argument: the Proposition 10 in [2] implies that the sub-sampled eigenvalues in our proposed regularizer would be much closer to the largest eigenvalues of the underlying FIM rather than the smallest ones, so that optimizing the upper bound tr(I) would behave quite differently than directly optimizing det(I). The argument is evidenced by a concentration bound.
>
> However, in practice, the bound is vacuously loose. Let us focus on the practical scenarios to do the analysis since otherwise, one would simply optimize det(I) directly instead of figuring out a tractable and effective proxy. The concentration bound shown as Proposition 10 has the $\kappa$ in the numerator and the $\sqrt{n}$ in the denominator. The former is the largest eigenvalues of the full FIM, and the latter is the square root of the batch size, making the bound already quite loose. Note that when we compute gradients in Algorithm 1, we do not compute them individually for each data point in the batch. Instead, we split the mini-batch into several sub-batches and compute the averaged gradients of the sub-batch. This has its own practical reason described at the end of page 6. And accordingly, the number of effective “batch size” used in approximating the tr(I) is reduced to a number normally smaller than 10. Together with the $2 \sqrt{2}$ in the numerator, the bound is indeed vacuous.
>
> Let us clarify and give a quick intuition for why our sub-sampled eigenvalues are not likely to be large. Since each gradient computed in approximating the tr(I) is the averaged gradient across a sub-batch, roughly speaking, the resulting “sub-sampled” Gram matrix over the averaged gradients has its spectral norm effectively reduced, thus alleviating the issue raised by the reviewer.
>
> Furthermore, in order to demonstrate that our regularizer indeed induces local minima which have smaller values according to our proposed metric (not merely smaller values by its upper bound), we compute our metric on local minima of similar training loss obtained with or without the regularizer. The numerical results are in Section 7.2.2.
>
> [1] Karakida, Ryo, Shotaro Akaho, and Shun-ichi Amari. "Universal statistics of fisher information in deep neural networks: mean field approach." arXiv preprint arXiv:1806.01316 (2018).
>
> [2] Rosasco, Lorenzo, Mikhail Belkin, and Ernesto De Vito. "On learning with integral operators." Journal of Machine Learning Research 11.Feb (2010): 905-934

---

> > ### Comment · AnonReviewer2 · 2019-11-13
> > **Current Response Not Convincing**
> >
> > Thanks for your reply. I agree there is a discrepancy about the definition of FIM initially; however, I do not find part (II) and (III) of your response convincing for the reasons below:
> >
> > =============================================================
> >
> > (A) The role of Information Theory, and confusions in Section 5.1
> >
> > In part (II) you did not address my concern, namely the FIA is defined using the expected FIM instead of the observed FIM, and it would be "confusing, irrelavent to the rest of the paper, and cannot be used to justify the work." To be specific, here is one place where the introduction of FIA created confusion:
> >
> > 1. Just above Sec 5.2, you defined $\gamma(w_0)$ as $\log|\mathcal{I}(w_0)|$, which is "undefined unless $|I(w_0)|\ne 0$". You then claims (Pennington and Worah, 2018) showed $I(w_0)$ is generally non-singular. However, at the second line on Page 3, Pennington and Worah stated that the matrix they studied "is equal to the Fisher information matrix of the model distribution with respect to its parameters". So it is clear they are studying the expected FIM $I_w$, not the observed FIM $I$ you claims.
> >
> > 2. You might argue that $w_0$ in Sec 5.1 only refers to the global minimum, as it is defined in Sec 4.1. However, you mentioned Theorem 1 applies to "*a* local minimum $w_0$", i.e. *any* local minimum; yet in the bound $\gamma(w_0)$ appeared. As $\gamma$ is only defined in Sec 5.1, to readers it would appear the discussions about the validity of $\gamma$ applies here, which cannot be true, as I've argued in point (1).
> > Besides, Theorem 1 must apply to any local minima, since it is about model parameter selection, not hyperparameter selection. Restricting it to the global minima would cut its link to the regularization method you developed below.
> >
> > 3. Now that $w_0$ may refer to any local minima, the statement in Sec 5.1 that "the last term [of FIA] becomes $ln V + ln \sqrt{|I(w_0)|}$" becomes misleading: It is only true for the global minimum where observed and expected FIM coincide.
> >
> > For this reasons I believe this section, as well as Sec 5.2, must be revised.
> >
> > =============================================================
> >
> > (B) Whether the underlying behavior of the proposed regularizer is as expected
> >
> > You claim
> > > Secondly, the reviewer argues that “det(I) focusing on the average eigenvalue yet tr(I) focusing on the few largest ones”. This is not the case since at a local minimum tr(I) is the L1 norm of the eigenvalues of I,
> >
> > Indeed tr(I) is the L1 norm of the eigenvalues. But my point has been that as the eigenvalues decay very quickly, thus the average is dominated by the first few ones: consider the global optima $w_0$, where the observed FIM and the expected coincide. As empirically shown in Fig 1 in (Karakida et al, AISTATS 2019), eigenvalue of the FIM decays exponentially. In this case, log(tr(I)) will be determined by the largest few values (a small and constant number of them, to be precise), while log(det(I)) would be dominated by the large number of small values.
> >
> > Another issue is that, as I've pointed out in (A), it appears that you do not have any justification about the validity of regularizer $log|\mathcal{I}(w_0)|$, at any $w_0$ that is not a global optima; your justification in Sec 5.1 only applies to $\log|\mathcal{I}_w(w_0)|$.
> >
> > For these reasons, I strongly believe the numerical experiment, as well as additional references justifying the use of $\log|\mathcal{I}|$ instead of $\log|\mathcal{I}_w|$, are needed.
> >
> > =============================================================
> >
> > (C)
> >
> > Finally, you claim there is no ambiguity in the notations chosen in Sec 5.1. In (A) I've given an example where ambiguity appears.

---

> > > ### Author Response · Authors · 2019-11-13
> > > **Quick question; just to confirm we understand the review correctly.**
> > >
> > > Thanks for the response. We are working on and will post our response and the revision of our paper.
> > >
> > > In the meantime, we are a little bit confused about the following request. For "I strongly believe the numerical experiment, as well as additional references justifying the use of $\log|\mathcal{I}|$ instead of  $\log|\mathcal{I}_w|$, are needed", what does $\log|\mathcal{I}|$ refer to? I assume the reviewer asks for the justification of the use of [X] instead of [Y] for the regularizer.

---

> > > ### Author Response · Authors · 2019-11-14
> > > **Response to reviewer #2 regarding part (II) & (III)**
> > >
> > > Thanks for the reply. We have updated the paper.
> > >
> > > Q1: The claims in (Pennington and Worah, 2018) that FIM is generally non-singular applies for expected FIM, not necessarily for observed FIM.
> > >
> > > Thanks for pointing this out. We have removed this in the updated version and support the non-singularity argument (i.e., our Assumption 1 is reasonable) in Section 5.1.
> > >
> > > Q2: FIA criterion only works for global minima and thus assuming global minima in the paper is a must.
> > >
> > > The FIA criterion does not require the local minimum to be globally optimal for the entire parameter space. When we introduce FIA criterion in Section 5.1, we state that it is for the model class of the local minimum (i.e., a statistical model incorporating all neural networks in the local minimum’s well-defined neighborhood), not for the entire parameter space. Otherwise, there is no comparison in the first place. By Assumption 1, a local minimum at our interest is indeed a unique global minimum in its model class.
> > >
> > > Furthermore, in the previous review, we stated that the local minima we care about are also assumed to be global minima. This assumption is not a requirement. We considered comparing global minima because this scenario is well-motivated, as explained in the beginning of Section 1.
> > >
> > > Q3: “Theorem 1 must apply to any local minima. Restricting it to the global minima would cut its link to the regularization method you developed below.”
> > >
> > > Indeed Theorem 1 applies to any local minima satisfying Assumption 1 & 2, as stated precisely in Theorem 1. We have mentioned in the answer above that global optimality is not a requirement.
> > >
> > > Q4: “ ‘the last term [of FIA] becomes’ is misleading”
> > >
> > > We believe this is a misinterpretation. We have clarified this in the updated version by changing the order we introduce our proposed metric and FIA. We meant to show the relationship between FIA and our metric, by no means to claim the terms are equivalent. As we have mentioned in the previous review, the FIA criterion is not used to derive or describe our proposed generalization bound.
> > >
> > > Q5: “It is only true for the global minimum where observed and expected FIM coincide.”
> > >
> > > Though irrelevant to our response above, we would like to kindly point out that this is factually inaccurate. The global optimality is neither a sufficient nor a necessary condition of “observed FIM coinciding expected FIM” unless the amount of training data goes to infinity. The unbiasedness of the maximum likelihood estimator does not indicate its efficiency.
> > >
> > > Q6: “the eigenvalues decay very quickly, thus the average is dominated by the first few ones”
> > >
> > > Thanks for clarifying this. As most modern network architectures are over-parameterized, we believe that in practice the difference between using tr(I) in the proposed regularizer instead of using the intractable det(I) is not critical. We understand the reviewer’s concern that large eigenvalues have a larger impact on tr(I) than the impact of that on det(I). However, as the reviewer has pointed out, Karakida et al, AISTATS 2019 demonstrates that the majority of the eigenvalue of the FIM is small with only a very few ones that can be large. For over-parameterized network where there are more parameters than training samples, whenever you compute the trace of FIM you actually compute the trace of FIM’s principal submatrices (see details in Section 5.3). By Theorem 3, the trace of the submatrix version is a “sub-sampled” version which is quite unlikely to have extremely large eigenvalues being “sampled” given that only a very small amount of the eigenvalues are large. The specific probability of picking such a large eigenvalue requires the study of extreme value theory of spectral density of the submatrices, which is beyond the scope of this paper. Furthermore, we apply gradient clipping in Algorithm 1 to restrict the effect, if any, of the extreme eigenvalues encountered in optimizing tr(I),
> > >
> > > Again, as we have shown in the previous review, the numerical results suggest that the generalization boost obtained from our regularizer can be attributed to what we expected -- the better local minima characterized by our proposed metric.
> > >
> > > Q7: “justification about the validity of regularizer not at a global optima”
> > >
> > > As answered previously, the global optimality is not a requirement in both theory and practice of our work.
> > >
> > > Q8: “There are ambiguity in the notations chosen in Sec 5.1”
> > >
> > > We have clarified the potential ambiguities in the updated version.

---

> > > > ### Comment · AnonReviewer2 · 2019-11-14
> > > > **Thanks for the reply; still there are misunderstanding and unaddressed concerns**
> > > >
> > > > It appears that there are still some misunderstanding (e.g. Q2 was never my question), as well as unaddressed concerns. Let me clarify.
> > > >
> > > >
> > > > (A) Relation between FIA and your regularized objective
> > > > =======================================================
> > > >
> > > > Let me re-state my concern: The FIA criterion is defined using the expected FIM; but at the end of Section 5.1.1, it is replaced with the observed FIM. This is only valid when the conditional distribution defined by the model, $p(y|x;w_0)$, coincides with the conditional data distribution $p_{data}(y|x)$ (or its smoothed version, in your setup).
> > > >
> > > > It is true that expected and observed FIM do not always coincide, even at global minima. However it is reasonable to consider the asymptotic setup, which is also what you mentioned in the previous draft. In this case, the replacement can only be valid when $w_0$ is a global minima in the entire parameter space, not only in the neighborhood $\mathcal{M}(w_0)$; in the finite sample case, it becomes even less clear when (or if) your replacement is valid or reasonable.
> > > >
> > > > In the revised version you explicitly mentioned the replacement of $\mathcal{I}_w$ with $\mathcal{I}$ (which is not because of any "asymptotic equivalence"). It is certainly helpful for clarification, but the replacement itself is still not well-justified, outside the global optimas that could potentially be justified by an asymptotic argument; so is the *connection* of your work to FIA.
> > > >
> > > >
> > > > (B) On the connection between log|I|/W and log(tr(I)/W)
> > > > =======================================================
> > > >
> > > > Unfortunately I cannot follow your argument in this part. Most importantly,
> > > >
> > > > > By Theorem 3, the trace of the submatrix version is a ``sub-sampled'' version which is quite unlikely to have extremely large eigenvalues
> > > >
> > > > When the eigenvalues vary in a wide range, the bounds in Theorem 3 merely becomes vacuous. It does not indicate the eigenvalue of the principal submatrices are uniformly (in any sense) distributed in that large bound.
> > > >
> > > > Furthermore, from the first equation in Page 6 you appear to be randomly subsampling a Gram matrix (of the neural tangent kernel). In this case standard concentration results (e.g. Proposition 10, Rosasco et al, 2010) imply the eigenvalue of your submatrix converges to the *largest* eigenvalues of the Gram matrix.
> > > >
> > > > > We understand the reviewer’s concern that large eigenvalues have a larger impact on tr(I)
> > > >
> > > > The equally important part is the tail that decays exponentially. It would dominate log(det(I)) and have vanishing impact on log(tr(I)). For this reason, one would expect the behavior of the two regularizers are sharply different.
> > > >
> > > > Finally, I can't follow the following sentence:
> > > >
> > > > > As most modern network architectures are over-parameterized, we believe that in practice the difference between using tr(I) in the proposed regularizer instead of using the intractable det(I) is not critical
> > > >
> > > > This is actually exacerbated by over-parameterization, as theoretical results like (Karakida et al) only work in the over-parameterized scheme.
> > > >
> > > >
> > > > Minor Points
> > > > ============
> > > >
> > > > * It is not clear from Section 4 that you focused on global optimas. The original words are
> > > > > Label smoothing enables us to assume a local minimum w0 (in this case, also a global minimum) of the training loss with [sum KL] = 0.
> > > > To me, it only appears that you are using label smoothing to make sure useful local minimas exist. Please consider revising.
> > > >
> > > > * I fail to see the need to mention the efficiency of MLE. Efficiency is about the asymptotic variance, not recovery of the true conditional distribution in any finite-sample case.
> > > >
> > > >
> > > > References
> > > > ==========
> > > >
> > > > Rosasco, Belkin and De Vito, On Learning with Integral Operators, JMLR 11 (2010)

---

> > > > > ### Author Response · Authors · 2019-11-15
> > > > > **Final response to reviewer #2**
> > > > >
> > > > > Dear reviewer #2,
> > > > >
> > > > > Q1: “Validation” of the connection between FIA and our proposed metric.
> > > > >
> > > > > We have updated the Section 5.1.1 again including adding a remark at the end to clarify the difference and connection between FIA and our metric. We would like to reiterate that the FIA criterion is NOT used in the paper to derive any theory or practice. The entire paper does NOT require any information-theoretic interpretation to be correct. The FIA criterion is stated as an important related work of our approach. We hope this time your concern will be resolved.
> > > > >
> > > > > Q2: “Proposition 10 in [1] implies that eigenvalue of the submatrix converges to the largest eigenvalues of the Gram matrix [and thus] the behavior of the two regularizers (directly optimizing det(I) vs. optimizing the upper bound tr(I)) are sharply different.”
> > > > >
> > > > > Let us first state the reviewer’s argument: the Proposition 10 in [1] implies that the sub-sampled eigenvalues in our proposed regularizer would be much closer to the largest eigenvalues of the underlying FIM rather than the smallest ones, so that optimizing the upper bound tr(I) would behave quite differently than directly optimizing det(I). The argument is evidenced by a concentration bound. However, in practice, the bound is vacuously loose. Let us focus on the practical scenarios to do the analysis since otherwise, one would simply optimize det(I) directly instead of figuring out a tractable and effective proxy. The concentration bound shown as Proposition 10 has the $\kappa$ in the numerator and the $\sqrt{n}$ in the denominator. The former is the largest eigenvalues of the full FIM, and the latter is the square root of the batch size, making the bound already quite loose. Note that when we compute gradients in Algorithm 1, we do not compute them individually for each data point in the batch. Instead, we split the mini-batch into several sub-batches and compute the averaged gradients of the sub-batch. This has its own practical reason described at the end of page 6. And accordingly, the number of effective “batch size” used in approximating the tr(I) is reduced to a number normally smaller than 10. Together with the $2 \sqrt{2}$ in the numerator, the bound is indeed vacuous.
> > > > >
> > > > > Let us clarify and give a quick intuition for why “sub-sampled eigenvalues are not likely to be large”. Since each gradient computed in approximating the tr(I) is the averaged gradient across a sub-batch, roughly speaking, the resulting “sub-sampled” Gram matrix over the averaged gradients has its spectral norm effectively reduced, thus alleviating the issue raised by the reviewer.
> > > > >
> > > > > Finally, we thank the reviewer for the time and effort during the week. We will try our best to update the revised version at the request of any further minor fixes..
> > > > >
> > > > > [1] Rosasco, Lorenzo, Mikhail Belkin, and Ernesto De Vito. "On learning with integral operators." Journal of Machine Learning Research 11.Feb (2010): 905-934

---

### Official Review · AnonReviewer3 · 2019-10-28
**Official Blind Review #3**

**Rating:** 3

**Review:**

This paper provides a metric to characterize local minima of deep network loss landscapes based on the Fisher information matrix of the model parameterized by the deep network. The authors connect the Fisher information to the curvature of the loss landscape (the loss considered is the negative loss likelihood) and obtain generalization bounds through PAC Bayes analysis. They further propose regularizing the training of deep networks using the local curvature of the loss as a regularizer. In the final experimental section of the paper, the relationship between the empirical measures and generalization is shown on a variety of networks.

This is an interesting paper, but I have a few concerns.

1. The information-theoretic measure that is proposed is essentially the (log) determinant of the hessian of the loss function.  If there are degenerate eigendirections (zero eigenvalues) then the proposed measure would not be able to distinguish between minima with different numbers of degenerate directions / same number of degenerate directions but different spectral norms of the hessians. If the authors contention is that there will be no zero eigenvalues, that suggests that local minima of deep networks are all strict, isolated minima, contrary to recent work on connected solutions (See Draxler et. al. 2018, Essentially No Barriers in Neural Network Energy Landscapes, ICML 2018).

2. I would like to see how the authors believe their measure deals with rescalings layer parameters in deep networks, ie the issue brought up by Dinh et. al. in "Sharp Minima can Generalize for Deep Networks" ICML 2017. While I can see that the log determinant is invariant, it is not clear that the proposed approximation will be invariant to rescaling of deep network layer parameters. If the parameters corresponding to the eigenvalues sampled in the approximation are rescaled, I believe the proposed measure will not be invariant.

3. The experiments regarding the local minima characterization are well constructed, though some details are missing such as how the authors decided that training had converged to a local minimum. As far as regularization based on the local curvature is concerned, I would like to see some more experiments that compare the proposed technique to adagrad/adam and other techniques that purport to condition the gradient based on local curvature. It would also be interesting to see whether the regularization indeed converges to flatter minima characterized by the proposed flatness measure. Since the claim is that the regularizer gets you flatter solutions, that information is important to decide whether the proposed technique is performing as advertised.

I am willing to update my score based on responses to these concerns.

**Experience Assessment:**

I have published one or two papers in this area.

**Review Assessment: Checking Correctness Of Derivations And Theory:**

I assessed the sensibility of the derivations and theory.

**Review Assessment: Checking Correctness Of Experiments:**

I assessed the sensibility of the experiments.

**Review Assessment: Thoroughness In Paper Reading:**

I read the paper thoroughly.

---

> ### Author Response · Authors · 2019-11-08
> **Response to reviewer #3**
>
> Dear reviewer3,
>
> Thank you for the constructive review.
>
> Q1: “There are degenerate eigendirections” or otherwise “local minima are all isolated”, which is “contrary to recent work [2] ...”
>
> Throughout our paper, we make the assumption that the local minima we care about are isolated, mentioned at the end of Section 5.1. We would like to update our paper to add a discussion on this subject, including the answer to this and the next question. This assumption is not necessarily contradictory to the argument that local minima are connected, as suggested by [2] that a relatively flat path exists between any pair of local minima of low training loss. We point out that the claim in [2] is not conclusive, that all the local minima are “perhaps best seen as” one connected component. In other words, the local minima can still be isolated.
>
> For state-of-the-art network architectures used in practice, the isolation assumption is often the fact. To be precise, this assumption is violated when the Hessian matrix at a local minimum is singular. Specifically, [3] summarizes three sources of the singularity: (i) due to a dead neuron, (ii) due to identical neurons, and (iii) linear dependence of the neurons. As well demonstrated in [3], network with skip connection (such as ResNet, WRN, and DenseNet used in our experiments) can effectively eliminate all the aforementioned singularity. There is another kind of singularity specifically for ReLU networks, which we will discuss next.
>
> Q2: In practice how can the proposed metric “deal with rescaling layer parameters in deep networks”, i.e., the rescaling issue described in [1]?
>
> In practice, the rescaling issue is not critical. There are three reasons:
> (I) This issue can only happen in neural networks equipped with scale-invariant activation functions, such as ReLU. Many state-of-the-art models use other activation functions such as ELU [7] that is not scale-invariant.
> (II) Even for ReLU networks, most modern DNNs are free of this issue, since they have normalization layers such as BatchNorm [8] applied before the activation. BatchNorm shifts all the inputs to the ReLU function, which is equivalent to shifting the ReLU function horizontally. The shifted ReLU is no longer scale-invariant. The ResNet, WRN, and DenseNet used in our experiments all fall into this category.
> (III) Due to the ubiquitous use of normal distribution based weights initialization scheme and the L2 regularization / weight decay, most of the local minima obtained by gradient-based learning algorithms have weights of a relatively small norm. Consequently, in practice, we will not compare two local minima essentially the same but have one as the rescaled version of the other with a much larger norm of the weights.
>
> In summary, the rescaling issue is another source of the singularity but only for networks equipped with scale-invariant activation functions. And in practice, it is effectively eliminated.
>
> Q3: “How the authors decided that training had converged to a local minimum” in Section 7.1?
>
> For the experiments of local minima characterization in Section 7.1, in all scenarios, we train the model for 200 epochs with an initial learning rate 0.1, divided by 10 when the training loss plateaus. Within each scenario, we find the final training loss very small and very similar across different models and the training accuracy essentially equal to 1, indicating the convergence.
>
> Q4: How is the proposed regularization method compared to “AdaGrad/Adam and other techniques that purport to condition the gradient based on local curvature”?
>
> Our regularizer aims to find better “flatter” minima to improve generalization whereas adaptive optimization methods such as AdaGrad and Adam try to boost up convergence, yet at the cost of generalizability. Recent works such as [4] and [5] show that adaptive methods generalize worse than SGD+Momentum. In specific, very similar to our setup, [5] demonstrates that SGD+Momentum consistently outperforms the others on ResNet and DenseNet for CIFAR-10 and CIFAR-100. Other approaches that also utilize local curvature, such as the Entropy-SGD [6] mentioned in Section 2, have empirical results rather preliminary compared to ours. Furthermore, as described in Algorithm 1, our proposed regularizer is not specific to a certain optimizer. We perform experiments with SGD+Momentum because it is chosen to be used in ResNet, WRN, and DenseNet, helping all of them achieve current or previous state-of-the-art results.

---

> > ### Author Response · Authors · 2019-11-08
> > **Response to reviewer #3 (cont.)**
> >
> > Q5: Whether the regularization “converges to flatter minima characterized by the proposed flatness measure”?
> >
> > Our regularizer essentially optimizes an upper bound of the proposed metric during training. As requested, for the following neural network architecture trained on CIFAR-10, we compute our metric on local minima of similar training loss obtained with or without the proposed regularizer. The following numerical results (each entry represents mean ± std among 5 runs) show that the resulting generalization boost indeed can be attributed to the “flatter” minima measured by our metric:
> >
> > =========================================================
> >   	                ResNet   	|     	     WRN           |         DenseNet
> > w\o reg       -979.3 ± 22.3          -737.6 ± 20.3              -850.3 ± 23.5
> > with reg     -1138.1 ± 11.0         -804.8 ± 18.7              -886.2 ± 20.5
> > =========================================================
> >
> >
> > [1] Dinh, Laurent, et al. "Sharp minima can generalize for deep nets." International Conference on Machine Learning, 2017.
> >
> > [2] Draxler, Felix, et al. "Essentially No Barriers in Neural Network Energy Landscape." International Conference on Machine Learning, 2018.
> >
> > [3] Orhan, A. Emin, and Xaq Pitkow. "Skip connections eliminate singularities." International Conference on Learning Representations, 2018.
> >
> > [4] Wilson, Ashia C., et al. "The marginal value of adaptive gradient methods in machine learning." Advances in Neural Information Processing Systems, 2017.
> >
> > [5] Keskar, Nitish Shirish, and Richard Socher. "Improving generalization performance by switching from adam to sgd." arXiv preprint arXiv:1712.07628 (2017).
> >
> > [6] Chaudhari, Pratik, et al. "Entropy-SGD: Biasing Gradient Descent Into Wide Valleys." International Conference on Learning Representations, 2017.
> >
> > [7] Clevert, Djork-Arné, Thomas Unterthiner, and Sepp Hochreiter. "Fast and accurate deep network learning by exponential linear units (elus)." International Conference on Learning Representations, 2016.
> >
> > [8] Ioffe, Sergey, and Christian Szegedy. "Batch Normalization: Accelerating Deep Network Training by Reducing Internal Covariate Shift." International Conference on Machine Learning. 2015.

---

> ### Author Response · Authors · 2019-11-15
> **Ping**
>
> We have addressed all the concerns in the previous response and updated our paper accordingly. Since tomorrow is the deadline for revision, we would like to ask if the reviewer #3 has any updated assessment or further concerns about our paper. Thanks a lot.

---

### Public Comment · ~Micah_Goldblum1 · 2019-11-08
**An Interesting Connection**

Hi Authors,
Thank you for your interesting paper.  I noticed that your work concerning generalization is related to our paper which visualizes the sharp-flat phenomenon, including experiments on realistic architectures, as an explanation for generalization.[1]  Please consider mentioning the relationship with our work in your next version.

[1] Huang, W. Ronny, et al. "Understanding Generalization through Visualizations." arXiv preprint arXiv:1906.03291 (2019).

---

> ### Author Response · Authors · 2019-11-13
> **Thanks**
>
> Thanks for your appreciation. We would definitely consider mentioning the relationship with this work in the next version of our paper.

---

### Author Response · Authors · 2019-11-14
**Updated version submitted**

Thanks for the reviewers. We have updated our paper including some of our responses. The new sections being added have their titles marked red.

---

### Decision · Program_Chairs · 2019-12-19

**Decision:**

Reject

**Comment:**

This paper proposes using the Fisher information matrix to characterize local minima of deep network loss landscapes to indicate generalizability of a local minimum. While the reviewers agree that this paper contains interesting ideas and its presentation has been substantially improved during the discussion period, there are still issues that remain unanswered, in particular between the main objective/claims and the presented evidence. The paper will benefit from a revision and resubmission to another venue.